# Do not trust what you trust: Miscalibration in Semi-supervised Learning

**Shambhavi Mishra**[1,2]                                    *shambhavi.mishra.1@etsmtl.net*
**Balamurali Murugesan**[1,2]                              *balamuralim.1993@gmail.com*
**Ismail Ben Ayed**[1,2]                                        *ismail.benayed@gmail.com*
**Marco Pedersoli**[1,2]                                        *marco.pedersoli@etsmtl.ca*
**Jose Dolz**[1,2]                                                    *jose.dolz@etsmtl.ca*

[1] *LIVIA, ÉTS Montréal, Canada*
[2] *International Laboratory on Learning Systems (ILLS),*
*McGILL - ETS - MILA - CNRS - Université Paris-Saclay - CentraleSupélec, Canada*

**Reviewed on OpenReview:** *https://openreview.net/forum?id=1WqLLYgBNt*

## Abstract

State-of-the-art semi-supervised learning (SSL) approaches rely on highly confident predictions to serve as pseudo-labels that guide the training on unlabeled samples. An inherent drawback of this strategy stems from the quality of the uncertainty estimates, as pseudo-labels are filtered only based on their degree of uncertainty, regardless of the correctness of their predictions. Thus, assessing and enhancing the uncertainty of network predictions is of paramount importance in the pseudo-labeling process. In this work, we empirically demonstrate that SSL methods based on pseudo-labels are significantly miscalibrated, and formally demonstrate the minimization of the min-entropy, a lower bound of the Shannon entropy, as a potential cause for miscalibration. To alleviate this issue, we integrate a simple penalty term, which enforces the logit distances of the predictions on unlabeled samples to remain low, preventing the network predictions to become overconfident. Comprehensive experiments on a variety of SSL image classification benchmarks demonstrate that the proposed solution systematically improves the calibration performance of relevant SSL models, while also enhancing their discriminative power, being an appealing addition to tackle SSL tasks. Code : https://github.com/ShambhaviCodes/miscalibration-ssl

## 1 Introduction

Deep learning models have significantly advanced the state-of-the-art across a myriad of tasks (Masana et al., 2022; Minaee et al., 2021). Nonetheless, their success has been often contingent on the availability of large amounts of labeled data. Having access to curated large training datasets, however, is not easy, and often involves a tremendous human labor, particularly in those domains where labeling data samples requires expertise, hindering the progress to address a broader span of real-world problems.

Semi-supervised learning (SSL) (Chapelle et al., 2006) mitigates the need for large labeled datasets by providing means of efficiently leveraging unlabeled samples, which are easier to obtain. This learning paradigm has led to a plethora of approaches, which can be mainly categorized into consistency regularization (Bachman et al., 2014; Laine & Aila, 2017) and pseudo-labeling (Lee et al., 2013; Xie et al., 2020b) methods. Indeed, state-of-the-art SSL approaches (Wang et al., 2023; Zhang et al., 2021; Sohn et al., 2020; Chen et al., 2023; Zheng et al., 2022) combine both strategies, obtaining promising results. The underlying idea of these approaches follows the low-density and smoothness assumptions in SSL (Chapelle & Zien, 2005). In particular, incorporating pseudo-labels from unlabeled data points into the training process aids the decision boundary to lie in low density regions. Furthermore, consistency regularization assumes that the same

unlabeled data point should yield the same pseudo-label regardless of the perturbations applied, implicitly capturing the underlying data manifold. As the model can produce very uncertain predictions for strongly perturbed samples, these techniques incorporate a threshold, either fixed (Sohn et al., 2020) or adaptive (Wang et al., 2023; Zhang et al., 2021), to only integrate very confident samples in the training loss. Thus, all samples whose predicted probabilities are highly confident are trusted by these methods as supervisory signals for subsequent steps, even when their predictions are wrong.

Despite being a standard practice, recent evidence (Chen et al., 2023) suggests that the amount of incorrect pseudo-labels integrated into the training is not negligible, potentially undermining the optimization process. Hence, given that the generated pseudo-labels play a significant role in the training of SSL models, producing accurate uncertainty estimates is of pivotal importance. Nevertheless, while we have observed a remarkable progress in their discriminative performance, little attention has been paid to studying, and improving, the calibration of SSL approaches. Motivated by these findings, in this work we address the critical yet under-explored issue of miscalibration in SSL, particularly for those methods based on pseudo-labeling. To this end, we select a set of relevant and recent strategies that build on pseudo-labels and consistency regularization (Sohn et al., 2020; Wang et al., 2023; Zhang et al., 2021) and empirically demonstrate that they are poorly calibrated. Furthermore, we explore the underlying causes of this issue and shed light about the potential reasons that produce overconfident pseudo-labeling SSL models. Last, inspired by these observations we propose a simple solution to tackle miscalibration in these models. Our contributions can be therefore summarized as follows:

1. We empirically demonstrate that state-of-the-art SSL approaches based on pseudo-labels are significantly miscalibrated. Through our analysis, we formally show that the cause of miscalibration is the minimization of a min-entropy term, a specific case of the Rényi family of entropies, on a considerably large proportion of unlabeled samples, which forces the model to yield overconfident predictions. Indeed, the ensuing gradients from this term strongly push the unlabeled samples to be highly confident since the beginning of the training, even if their class predictions are incorrect. This results in large logit magnitudes, a phenomenon known to cause miscalibration.

2. Based on our observations, we propose to use a simple solution that refrains the model from pushing unlabeled samples towards very confident regions, improving the calibration of pseudo-labeling SSL methods. More concretely, we add a penalty term on a dominant set of unlabeled samples, which enforces logit distances to remain low, alleviating the miscalibration issue.

3. Through a comprehensive set of experiments, we empirically demonstrate that the proposed approach consistently improves the uncertainty estimates of a set of very relevant and recent state-of-the-art pseudo-labeling SSL approaches on popular benchmarks, in both standard and long-tailed classification tasks. In addition, in most cases, the proposed solution further improves their discriminative performance.

## 2 Related work

### 2.1 Semi-supervised learning

Prevailing semi-supervised learning (SSL) approaches heavily rely on the concept of pseudo-labels (Lee et al., 2013; Shi et al., 2018) and consistency regularization (Bachman et al., 2014; Laine & Aila, 2017; Sajjadi et al., 2016; Tarvainen & Valpola, 2017; Miyato et al., 2018), where labels are dynamically generated for unlabeled data throughout the training process. Essentially, these methods exploit the role of perturbations, by stochastically perturbing the unlabeled images and enforcing consistency across their predictions. This consistency is achieved by a pseudo-supervised loss, where the predictions over the strong perturbations are supervised by the pseudo-labels obtained from the weak perturbations (Xie et al., 2020a; Wang et al., 2023; Zhang et al., 2021; Chen et al., 2023; Zheng et al., 2022; Yang et al., 2023; Xu et al., 2021; Zheng et al., 2022). This paradigm to use artificial labels facilitates the integration of unlabeled data into the learning process, thereby augmenting the training set and improving the model ability to generalize. To avoid introducing noise in the pseudo-supervision process, these approaches retain a given pseudo-label only if the model

assigns a high probability to one of the possible classes. This strategy effectively harnesses the information from unlabeled data by leveraging the network confidence in assigning pseudo labels, enabling the model to access the valuable knowledge encapsulated within these unlabeled samples. Thus, the main differences across the different approaches based on pseudo-labels lie on the mechanism introduced to select confident samples. For example, FixMatch (Sohn et al., 2020) and ShrinkMatch (Yang et al., 2023) employ a fixed threshold, whereas Dash (Xu et al., 2021) proposes a dynamically growing threshold. Other approaches, such as FlexMatch (Zhang et al., 2021) and FreeMatch (Wang et al., 2023), integrate class-adaptative thresholds, considering a larger amount of unlabeled data which is otherwise ignored, especially at the early stage of the training process.

*Limitations of pseudo-labeling SSL from a calibration standpoint.* Although these methods enhance the discriminative power of deep models, their calibration has been significantly overlooked, lacking of principled strategies to simultaneously improve the classification performance while maintaining the quality of the uncertainty estimates. As pseudo-labeling state-of-the-art SSL approaches trust highly confident artificial labels derived from unlabeled samples, understanding how this confidence is assigned, and ensure its accuracy, is of paramount importance. In this context, (Rizve et al., 2021) addresses the challenge of selecting reliable pseudo-labels by incorporating uncertainty estimates into the pseudo-labeling process. Very recently, BAM (Loh et al., 2023) studied miscalibration in SSL, and proposed to replace the last layer of a neural network by a Bayesian layer. Nevertheless, the source of miscalibration was not explored, and in-depth empirical results were not reported.

## 2.2 Calibration

Recent evidence (Guo et al., 2017; Müller et al., 2019; Mukhoti et al., 2020) has shown that deep networks are prone to make overconfident predictions due to miscalibrated output probabilities. This emerges as a byproduct of minimizing the prevalent cross-entropy loss, which occurs when the softmax predictions for all training samples fully match the ground-truth labels, and thus the entropy of output probabilities is encouraged to be zero. To mitigate the miscalibration issue, and to better estimate the predictive uncertainty of deterministic models, two main families of approaches have emerged recently: *post-processing* (Guo et al., 2017; Ding et al., 2021; Tomani et al., 2021) and *learning* (Pereyra et al., 2017; Müller et al., 2019; Mukhoti et al., 2020; Liu et al., 2022; Cheng & Vasconcelos, 2022; Liu et al., 2023; Noh et al., 2023; Murugesan et al., 2023; Larrazabal et al., 2023; Park et al., 2023) approaches. Among the post-processing strategies, Temperature Scaling (TS) (Guo et al., 2017) has been a popular alternative, which manipulates logit outputs monotonely, by applying a single scalar temperature parameter. This idea is further extended in (Ding et al., 2021), where a local TS per pixel is provided by a regression neural network. Nevertheless, despite the simplicity of these methods, learning approaches have arisen as a more powerful choice, as in this scenario the model adapts to calibration requirements alongside its primary learning objectives, optimizing both aspects simultaneously. Initial attempts integrated learning objectives that maximize the entropy of the network softmax predictions either explicitly (Pereyra et al., 2017; Larrazabal et al., 2023), or implicitly (Mukhoti et al., 2020; Müller et al., 2019; Cheng & Vasconcelos, 2022). In order to alleviate the non-informative nature of simply maximizing the entropy of the softmax predictions, recent work (Liu et al., 2022; 2023) has presented a generalized inequality constraint, which penalizes logits distances larger than a pre-defined margin.

# 3 Semi-supervised learning and calibration

## 3.1 Problem statement

In the semi-supervised learning scenario, the training dataset is composed of labeled and unlabeled data points. In this setting, let $\mathcal{D}_L = \{(\mathbf{x}_i, \mathbf{y}_i)\}_i^{N_L}$ be the labeled dataset and $\mathcal{D}_U = \{\mathbf{x}_i\}_i^{N_U}$ the unlabeled dataset, where $N_L$ and $N_U$ represent the number of labeled and unlabeled samples, respectively, and $N_L << N_U$. Furthermore, $\mathbf{x}_i \in \mathbb{R}^d$ is a $d$-dimensional training sample, with $\mathbf{y}_i \in \{0,1\}^K$ its associated ground truth (only for labeled data points, i.e., $\mathbf{x}_i \in \mathcal{D}_L$) that assigns one of the $K$ classes to the sample. The objective is, given a batch of labeled and unlabeled samples, to find an optimal set of parameters of a deterministic

function, e.g., a neural network, parameterized by $\boldsymbol{\theta}$, by using a compounded loss including a labeled and an unlabeled term. Given an input image $\mathbf{x}_i$, the network will generate a vector of logits $f_{\boldsymbol{\theta}}(\mathbf{x}_i) = \mathbf{l}_i \in \mathbb{R}^K$, which can be converted to probabilities with the softmax function.

**Supervised loss.** The supervised objective is typically formulated as a standard cross-entropy $\mathcal{H}$ between the one-hot encoded labels $\mathbf{y}_i$ and the corresponding softmax predictions $\mathbf{p}(\mathbf{y}|\mathbf{x_i}) \in [0,1]^K$ of labeled samples:

$$\mathcal{L}_S = \mathcal{H}(\mathbf{y}_i, \mathbf{p}(\mathbf{y}|\mathbf{x_i})) = -\sum_{i \in \mathcal{D}_L} \mathbf{y}_i \log \mathbf{p}(\mathbf{y}|\mathbf{x_i}) \tag{1}$$

**Unsupervised loss.** Most modern SSL approaches adopt a consistency regularization strategy based on pseudo-labeling for the unsupervised objective. To this end, the same image $\mathbf{x}_i$ follows a set of weak and strong augmentations, denoted as $\omega(\cdot)$ and $\Omega(\cdot)$, respectively. Thus, a pseudo-cross-entropy term on the unlabeled training dataset can be formulated as:

$$\mathcal{L}_U = -\sum_{i \in \mathcal{D}_U} \tilde{\mathbf{y}}_i \log \mathbf{p}(\mathbf{y}|\Omega(\mathbf{x}_i)) \tag{2}$$

where $\tilde{\mathbf{y}}_i$ is the one-hot encoding of the $\arg\max$ of the softmax probabilities for the weak augmented version, i.e., $\arg\max(\mathbf{p}(\mathbf{y}|\omega(\mathbf{x}_i)))$. To avoid that samples with high uncertainty, and possibly incorrect predictions, intervene in the optimization of the term in eq. (2), a common strategy is to retain only discrete pseudo-labels whose largest class probability fall above a predefined threshold (Lee et al., 2013; Wang et al., 2023; Sohn et al., 2020; Zhang et al., 2021). This results in the following objective:

$$\mathcal{L}_U = -\sum_{i \in \mathcal{D}_U} \mathbb{1}(\max(\mathbf{p}(\mathbf{y}|\omega(\mathbf{x}_i))) \geq \tau) \tilde{\mathbf{y}}_i \log \mathbf{p}(\mathbf{y}|\Omega(\mathbf{x}_i)) \tag{3}$$

where $\tau$ is the predefined threshold. Note that all loss terms are normalized by the cardinality of each set, which we omit for simplicity.

## 3.2 Revisiting the calibration of semi-supervised models

We now introduce a series of observations revealing several intrinsic properties of semi-supervised methods built up on pseudo-labels generation, which allows us to motivate a calibration technique tailored for pseudo-label based SSL.

**Observation 1. Semi-supervised learning degrades the calibration performance.** An important body of literature on SSL relies on pseudo-labeling to leverage the large amount of unlabeled samples. To achieve this, a very common strategy is to generate weak augmentations of each unlabeled image, whose predictions serve as supervision for their strong augmentation counterpart, as presented in eq. (3). Pseudo-labeling (Lee et al., 2013) is indeed closely related to entropy regularization (Grandvalet & Bengio, 2004), which favors a low-density separation between classes, a commonly assumed prior for semi-supervised learning. While minimizing the entropy of the predictions can actually improve the discriminative performance of neural networks, it inherently favours overconfident predictions, which is one of the main causes of miscalibration. Figure 1 brings empirical evidence about this observation, where we can observe that *despite bringing performance gains, in terms of accuracy, pseudo-label based SSL methods naturally degrade the calibration properties of a supervised baseline* trained with a few labeled samples.

**Observation 2. Pseudo-labeling in SSL indeed minimizes min-entropy on unlabeled points.** The use of a hard label makes pseudo-labeling closely related to entropy minimization (Grandvalet & Bengio, 2004). Nevertheless, the different transformations that unlabeled images follow in modern SSL methods produce different probability distributions for the weak and strong versions of the same image, where the

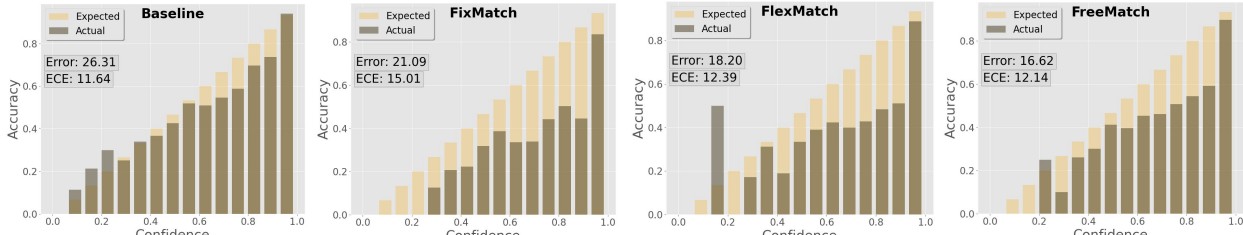

Figure 1: **Observation 1.** Reliability plots for a baseline supervised model (trained with eq. (1)) and three representative SSL approaches (trained with eq. (4)) on CIFAR-100. These plots empirically highlight the calibration degradation observed when training with the standard unsupervised loss, despite the gains achieved in discrimination.

predictions of the former are used to correct the predictions of the later. Note that this is slightly different from the traditional pseudo-labeling approaches, where the same image is used for assigning the pseudo-label and updating the predictive model. Thus, albeit they are related, the minimization of entropy cannot be attributed as the cause of miscalibration. Motivated by this, we explore in this section the implications of the standard SSL unlabeled loss based on pseudo-labels and its effect on network calibration. The common learning objective is composed of two terms: the first one is the standard cross-entropy (CE) on labeled samples (eq. (1)), while the second term is a CE between pseudo-labels obtained from weak augmentations and the predictions of their strong augmented counterparts (eq. (3)):

$$\mathcal{L}_T = \underbrace{-\sum_{i \in \mathcal{D}_L} \mathbf{y}_i \log \mathbf{s}_i}_{\text{CE on labeled samples}} \underbrace{-\sum_{i \in \mathcal{D}_U} \tilde{\mathbf{y}}_i^w \log \mathbf{s}_i^s}_{\text{Pseudo-CE on } \mathcal{D}_U}, \tag{4}$$

where $\mathbf{s}_i = \mathbf{p}(\mathbf{y}|\mathbf{x_i})$ is used for simplicity, and the superscripts $w$ and $s$ denote weak and strong transformations, respectively. We now split the unlabeled dataset into $\mathcal{D}_{\mathcal{U}'}$, which contains the unlabeled samples whose predicted class from weak and strong augmentations are different, i.e., $\arg\max(\mathbf{s}_i^w) \neq \arg\max(\mathbf{s}_i^s)$ and $\mathcal{D}_{\mathcal{U}''}$, containing the samples whose predicted class from weak and strong augmentations are the same. Thus, we can decompose the right-hand term in eq. (4) into two terms, one acting over $\mathcal{D}_{\mathcal{U}'}$ and one over $\mathcal{D}_{\mathcal{U}''}$:

$$\mathcal{L}_T = \underbrace{-\sum_{i \in \mathcal{D}_L} \mathbf{y}_i \log \mathbf{s}_i}_{\text{CE on labeled samples}} \underbrace{-\sum_{i \in \mathcal{D}_{U'}} \tilde{\mathbf{y}}_i^w \log \mathbf{s}_i^s}_{\text{Pseudo-CE on } \mathcal{D}_{U'}} - \sum_{i \in \mathcal{D}_{U''}} \tilde{\mathbf{y}}_i^w \log \mathbf{s}_i^s \tag{5}$$

In the above equation, the second term can be seen as a pseudo cross-entropy on $\mathcal{D}_{U'}$. Furthermore, as $\tilde{\mathbf{y}}_i^w$ is equal to the one-hot vector from $\arg\max(\mathbf{s}_i^s)$ on samples from $\mathcal{D}_{U''}$, the last term is equivalent to the min-entropy[1]:

$$\mathcal{L}_T = \underbrace{-\sum_{i \in \mathcal{D}_L} \mathbf{y}_i \log \mathbf{s}_i}_{\text{CE on labeled samples}} \underbrace{-\sum_{i \in \mathcal{D}_{U'}} \tilde{\mathbf{y}}_i^w \log \mathbf{s}_i^s}_{\text{Pseudo-CE on } \mathcal{D}_{U'}} \underbrace{-\sum_{i \in \mathcal{D}_{U''}} \log(\max_k \mathbf{s}_{i,k}^s)}_{\text{min-entropy on } \mathcal{D}_{U''}} \tag{6}$$

As shown in Figure 2 in the case of a two-class distribution $(p, 1-p)$, min-entropy is a lower bound of the Shannon Entropy, which has several implications in network miscalibration. In particular, while both

---

[1]Pseudo-label $\hat{y}_{i,c}^w = 1$ if $\mathbf{s}_{i,c}^s = \max_k \mathbf{s}_{i,k}^s$, and 0 otherwise.

Shannon Entropy and min-entropy reach their minimum at the vertices of the simplex, i.e., when $p = 0$ or $p = 1$ (*left*), the dynamics of the gradients are different (*middle*). More concretely, in the case of the Shannon entropy, the gradients of low-confidence predictions at the middle of the simplex are small and, therefore, dominated by the other terms at the beginning of training. In contrast, by minimizing the min-entropy, the inaccuracies resulting from uncertain predictions are reinforced, i.e., pushed towards the simplex vertices, yielding early errors in the predictions, which are hardly recoverable, and potentially misleading the training process.

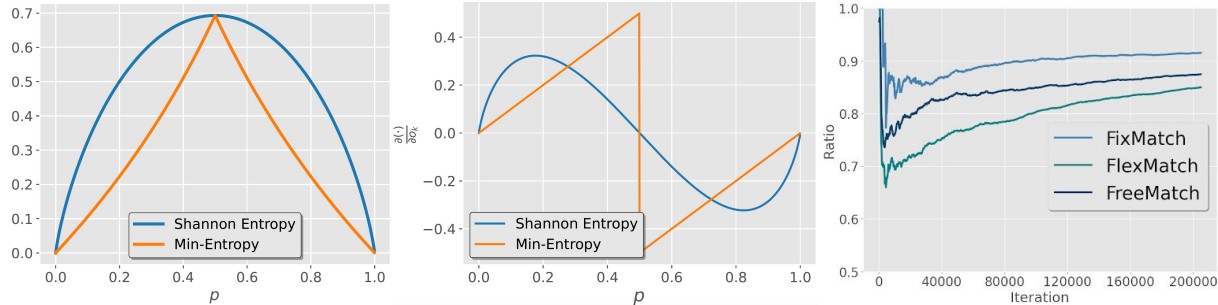

Figure 2: **Observation 2. (Left)** The unsupervised term in pseudo-label SSL is equivalent to a pseudo cross-entropy on a small subset of unlabeled samples and a pseudo-entropy regularization loss that minimizes the min-entropy (a lower bound of the Shannon Entropy) of the predictions from most unlabeled samples. **(Middle)** Compared to the Shannon Entropy, the min-entropy is more aggressive in the gradient dynamics, particularly at the beginning of the training, when most predictions are uncertain. **(Right)** Ratio of samples with same hard prediction for weak and strong augmentations that were above the selection threshold of three relevant SSL methods.

Furthermore, we empirically observed that the amount of samples where the $\arg\max$ of the predictions from weak and strongly augmented versions was the same, i.e., $\mathcal{D}_{U''}$, was significantly larger than those with different predictions, i.e., $\mathcal{D}_{U'}$ (Figure 2, *right*). In particular, as shown in this figure, and after some iterations, **more than 80% of unlabeled samples included in the training share the same pseudo-label between weak and strong annotations,** regardless of the approach analyzed. Thus, based on our observations we argue that *the training of SSL methods based on pseudo-labels is equivalent to having a supervised term coupled with a pseudo cross-entropy on a small subset of unlabeled samples and a pseudo-entropy regularization loss that minimizes the min-entropy of the predictions from most unlabeled samples.* This means that, while implicitly, or explicitly, minimizing the Shannon entropy on the network softmax predictions is known to cause miscalibration (Guo et al., 2017), employing the min-entropy aggravates the problem, which explains why SSL methods based on pseudo-labels are not well calibrated, particularly compared to a simple supervised baseline (Figure 1).

**Observation 3. Pseudo-label SSL techniques produce highly overlapped logit distributions, with large logit magnitudes and distances.** A direct implication of **Observation 2** is that softmax predictions in pseudo-labeling SSL methods become highly confident, which translates into larger logit values compared to a supervised baseline. As a result, even when a predicted category is incorrect, the network will still express a high level of certainty in its prediction. Take for instance the example shown in Figure 3, which depicts the logit value distributions for the samples belonging to class 5. We can observe that, in the case of the supervised baseline, the maximum logit values for incorrect predicted classes is at $\approx 7.5$ (classes 6 and 7) and around 6.5 (class 8). In contrast, when adding unlabeled samples in the form of pseudo-labels (e.g., in FixMatch (Sohn et al., 2020)), the maximum logit values for incorrect classes increase to nearly 12.5. This means that, while both models yield incorrect predictions, these are highly confident in the SSL methods. Furthermore, the logit range in the supervised baseline goes from around -7 to 10, whereas FreeMatch produces logits in the range from -10 to 15, which will result in highest probability scores for the predicted category. From these observations, we argue that *a well-calibrated SSL model should decrease the magnitude of logits associated with incorrect classes, as well as their total logits range, thereby decreasing*

*confidence in erroneous predictions, while simultaneously preserving high values in logits corresponding to the target class.*

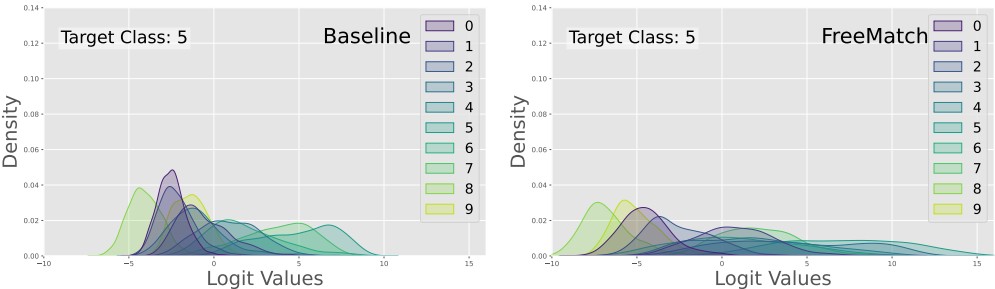

Figure 3: **Observation 3.** These plots depict the Kernel Density Estimation of the logit distributions obtained by *(left)* the supervised baseline trained with eq. (1) and *(right)* FreeMatch on STL-10, for the samples belonging to class 5. We can observe that, even for non-target classes (k≠5), the logit magnitudes in FreeMatch are larger, which translates to higher overconfidence in both correct and incorrect predictions. We select STL-10 due to its number of classes (10 *vs.* 100 in CIFAR-100).

## 4 Our solution

Based on our findings, especially in **Observation 3**, it becomes evident that an effective strategy for addressing miscalibration within the SSL scenario involves controlling the magnitude of predicted logits for unlabeled samples. Furthermore, from **Observation 2** we can derive that the underlying mechanism magnifying the miscalibration issue stems from the hidden min-entropy term on the data points in $\mathcal{D}_{U''}$, which represents the majority of unlabeled samples. Thus, we resort to an inequality constraint that imposes a controllable margin on the logit distances of predictions in samples from $\mathcal{D}_{U''}$. This constraint, which draws inspiration from (Liu et al., 2022), takes the following form $\mathbf{d(l)} \leq \mathbf{m}$, where $\mathbf{d(l)} = (\max_j(l_j) - l_k)_{1 \leq k \leq K} \in \mathbb{R}^K$ represents the vector of logit distances between the winner class and the rest (only for samples in $\mathcal{D}_{U''}$) and $\mathbf{m}$ a $K$-dimensional vector defining the margin values, with all elements equal to $m \in \mathbb{R}_{++}$. Note that while (Liu et al., 2022) was proposed in the fully supervised learning scenario, in this work we only enforce the constraint on a subset of samples, which is motivated by the observations presented in 3.2. Furthermore, this choice is also supported empirically in the ablation studies presented in the experiments.

By integrating this inequality constraint on the logit distances, training becomes a constrained problem, whose objective can formally defined as:

$$\begin{aligned} \text{minimize} \quad & \mathcal{L}_{\mathrm{T}} \\ \text{subject to} \quad & \mathbf{d(l)} \leq \mathbf{m} \qquad \mathbf{m} \in \mathbb{R}_{++}, \forall \mathbf{x}_i \in \mathcal{D}_{U''}. \end{aligned} \tag{7}$$

The above constrained problem in eq. (7) can be approximated by penalty-based optimization method, transforming the formulation into an unconstrained problem by using a simple ReLU function:

$$\min_{\boldsymbol{\theta}} \quad \mathcal{L}_{\mathrm{T}} + \lambda \sum_{i \in \mathcal{D}_{U''}} \sum_k \max(0, \max_j(l_{i,j}) - l_{i,k} - m_k) \tag{8}$$

where the second term, i.e., the non-linear ReLU penalty, prevents logit distances from exceeding a predetermined margin $m$, and $\lambda \in \mathbb{R}_+$ is a blending hyperparameter which controls the contribution of the CE loss and the corresponding penalty. The intuition behind this penalty term is simple. For the *winner* logits where the distance with the remaining logits is above the margin $m$, a gradient will be back-propagated to enforce those values to decrease. As a result, the whole logit magnitudes will decrease, potentially alleviating the miscalibration issue in the set of unlabeled samples $\mathcal{L}_{U''}$, which dominate the SSL training.

# 5 Experiments

**Datasets.** We resort to the recent Unified Semi-supervised Learning Benchmark for Classification (USB) (Wang et al., 2022), which compiles a diverse and challenging benchmark across several datasets. In particular, we focus on three popular datasets: **CIFAR-100** (Krizhevsky & Hinton, 2010), which has significant value as a standard for fine-grained image classification due to its wide range of classes and detailed object distinctions; **STL-10** (Coates et al., 2011), which is widely recognized for its limited sample size and extensive collection of unlabeled data, rendering it a challenging scenario of special significance in the context of SSL; and **EuroSAT** (Helber et al., 2019), containing 10 unique fine-grained categories related to earth observation and satellite imagery analysis, and important challenges such as high variability and imbalance classes. Last, we also conduct further experiments in the long-tailed version of CIFAR-100.

**Architectures.** We have prioritized Vision Transformers (ViT) over Convolutional Neural Networks (CNNs), for three main reasons related to discriminative performance, quality of uncertainty estimates, generalization and transfer learning capabilities. First, the emergence of ViTs has proven these models to outperform their CNNs counterparts (Raghu et al., 2021; Cai et al., 2022). Second, from a calibration standpoint, ViTs have also shown to be better calibrated than CNNs (Minderer et al., 2021; Pinto et al., 2021). Hence, due to their superior discriminative and calibration performance, they pose a more challenging scenario to evaluate the effectiveness of the proposed strategy. And last, the fine-tuning capabilities of ViTs enable effective transfer learning across diverse visual tasks and datasets. This capability is particularly advantageous in scenarios where labeled data is scarce, as it allows leveraging pre-trained representations learned from large-scale datasets, significantly minimizing the amount of training iterations while maintaining consistent performance[2]. More concretely, we employ a ViTSmall (Gani et al., 2022) with a patch size of 2 and an image size of 32 for CIFAR-100 and EuroSAT, in accordance with the standard in USB, and a ViT-Small with an image size of 96 for STL10.

We use same settings for all models and benchmarks to provide a fair comparison.

**Training, evaluation protocol and metrics.** While several works use different amounts of labeled data in their experiments, we perform due diligence and follow the settings proposed in USB (Wang et al., 2022) for training. For each method and configuration, we perform three runs with different seeds, select the best checkpoint and report their mean and standard deviation, following the literature. We report error rates for the accuracy performance and the expected calibration error (ECE), following the literature in calibration of supervised models. Implementation details are discussed in Appendix.

## 5.1 Results

**Main results**. The proposed strategy is model agnostic, and can be integrated on top of any SSL approach based on pseudo-labels, enabling substantial flexibility. For the empirical evaluation, we selected three popular and relevant approaches that resort to hard pseudo-labels: FixMatch (Sohn et al., 2020), FlexMatch (Zhang et al., 2021) and FreeMatch (Wang et al., 2023) and assess the impact of adding our simple solution during training. *Discriminative performance (table 1):* we observe that in 16 out of the 18 different settings, adding the penalty in eq. (8) brings improvement gains compared to the original versions of each method, which are only trained with eq. (4). Note that these gains are sometimes substantial, improving the original method by up to 4% (e.g., FixMatch in CIFAR-100(200) and EuroSAT(20) or FlexMatch in EuroSAT(20)). Furthermore, the three approaches combined with our penalty achieve very competitive performance compared to existing SSL literature, typically yielding state-of-the-art results. *Calibration performance (table 2):* Similarly, including the penalty term systematically enhances the calibration performance of the three analyzed approaches, whose improvements are typically significant (up to 6-7% in several cases).

An interesting observation is that, MixMatch (Berthelot et al., 2019), a consistency regularization based approach, yields surprisingly well calibrated models. Indeed, MixMatch has several components that have shown to improve calibration, such as ensembling predictions and MixUp, which may explain the obtained values. Nevertheless, it is noteworthy to mention that the discriminative performance compared to our

---

[2]Training FreeMatch on CIFAR100 with 400 labeled samples goes from 12 days (WideResNet from scratch) to 10 hours (ViTSmall) in an NVIDIA V100-32G GPU.

Table 1: **Classification performance (error rate (%)).** Arrows indicate whether our modified version improves ($\downarrow$) or deteriorates ($\uparrow$) the performance. Best overall performance in bold and best across pseudo-label SSL approaches underlined.

| Dataset | CIFAR-100 | | EuroSAT | | STL-10 | |
|---|---|---|---|---|---|---|
| # Labeled samples | 200 | 400 | 20 | 40 | 40 | 100 |
| *Only consistency regularization* | | | | | | |
| MixMatch[NeurIPS'19] | $37.68_{\pm 2.66}$ | $26.84_{\pm 1.06}$ | $28.77_{\pm 10.40}$ | $14.88_{\pm 2.07}$ | $25.19_{\pm 2.05}$ | $11.37_{\pm 1.49}$ |
| Dash [ICML'21] | $28.51_{\pm 2.91}$ | $19.54_{\pm 1.20}$ | $10.05_{\pm 8.15}$ | $6.83_{\pm 3.24}$ | $18.30_{\pm 4.58}$ | $8.74_{\pm 2.13}$ |
| AdaMatch [ICLR'22] | $\mathbf{19.26}_{\pm 1.83}$ | $17.13_{\pm 0.92}$ | $12.01_{\pm 4.16}$ | $6.07_{\pm 2.26}$ | $13.31_{\pm 3.75}$ | $8.14_{\pm 1.48}$ |
| DeFixMatch [ICLR'23] | $30.44_{\pm 0.82}$ | $20.93_{\pm 1.42}$ | $14.27_{\pm 9.05}$ | $5.42_{\pm 2.69}$ | $25.36_{\pm 4.40}$ | $10.97_{\pm 1.75}$ |
| *Pseudo-labeling* | | | | | | |
| FixMatch[NeurIPS'20] | $31.28_{\pm 1.58}$ | $19.42_{\pm 1.56}$ | $11.88_{\pm 6.32}$ | $6.64_{\pm 5.03}$ | $16.13_{\pm 2.36}$ | $8.06_{\pm 2.15}$ |
| FixMatch + Ours | $27.57_{\pm 1.49}\downarrow$ | $18.48_{\pm 1.65}\downarrow$ | $7.19_{\pm 4.83}\downarrow$ | $5.02_{\pm 2.24}\downarrow$ | $17.55_{\pm 4.00}\uparrow$ | $7.96_{\pm 1.64}\downarrow$ |
| FlexMatch[NeurIPS'21] | $28.27_{\pm 0.59}$ | $17.61_{\pm 0.51}$ | $7.89_{\pm 3.06}$ | $7.13_{\pm 1.23}$ | $13.34_{\pm 1.63}$ | $8.35_{\pm 1.24}$ |
| FlexMatch + Ours | $26.49_{\pm 0.52}\downarrow$ | $18.15_{\pm 0.47}\uparrow$ | $\mathbf{3.69}_{\pm 0.81}\downarrow$ | $5.00_{\pm 0.98}\downarrow$ | $\underline{\mathbf{12.87}}_{\pm 4.32}\downarrow$ | $\underline{\mathbf{7.53}}_{\pm 1.32}\downarrow$ |
| FreeMatch[ICLR'23] | $23.92_{\pm 2.02}$ | $16.18_{\pm 0.38}$ | $4.74_{\pm 1.77}$ | $4.48_{\pm 0.73}$ | $14.88_{\pm 0.72}$ | $8.83_{\pm 0.14}$ |
| FreeMatch + Ours | $\underline{21.36}_{\pm 1.62}\downarrow$ | $\underline{\mathbf{16.09}}_{\pm 0.80}\downarrow$ | $4.30_{\pm 1.46}\downarrow$ | $\mathbf{3.50}_{\pm 0.70}\downarrow$ | $13.18_{\pm 1.61}\downarrow$ | $8.57_{\pm 1.05}\downarrow$ |

modified versions of SSL methods is strikingly lower, with differences typically going from 10% to 24%, failing to achieve a good compromise between accuracy and calibration.

Table 2: **Calibration performance (ECE).** Arrows indicate whether our modified version improves ($\downarrow$) or deteriorates ($\uparrow$) the performance. Best overall performance in bold, whereas best across pseudo-label SSL approaches is underlined.

| Dataset | CIFAR-100 | | EuroSAT | | STL-10 | |
|---|---|---|---|---|---|---|
| # Labeled samples | 200 | 400 | 20 | 40 | 40 | 100 |
| *Only consistency regularization* | | | | | | |
| MixMatch[NeurIPS'19] | $\mathbf{8.13}_{\pm 2.16}$ | $\mathbf{7.19}_{\pm 2.31}$ | $9.27_{\pm 3.68}$ | $3.75_{\pm 3.30}$ | $\mathbf{3.42}_{\pm 1.77}$ | $5.89_{\pm 1.51}$ |
| Dash [ICML'21] | $22.23_{\pm 2.85}$ | $13.20_{\pm 1.07}$ | $7.09_{\pm 6.65}$ | $4.24_{\pm 2.22}$ | $11.23_{\pm 2.33}$ | $5.23_{\pm 1.87}$ |
| AdaMatch [ICLR'22] | $12.96_{\pm 1.76}$ | $11.17_{\pm 0.72}$ | $8.55_{\pm 4.83}$ | $2.66_{\pm 0.67}$ | $8.80_{\pm 3.01}$ | $4.96_{\pm 1.25}$ |
| DeFixMatch [ICLR'23] | $24.54_{\pm 1.37}$ | $14.89_{\pm 0.97}$ | $10.46_{\pm 8.71}$ | $2.90_{\pm 1.29}$ | $12.89_{\pm 1.73}$ | $6.62_{\pm 1.87}$ |
| *Pseudo-labeling* | | | | | | |
| FixMatch[NeurIPS'20] | $27.77_{\pm 1.49}$ | $13.45_{\pm 1.45}$ | $8.36_{\pm 5.29}$ | $4.72_{\pm 4.45}$ | $10.27_{\pm 2.4}$ | $5.83_{\pm 2.25}$ |
| FixMatch + Ours | $21.56_{\pm 1.32}\downarrow$ | $12.12_{\pm 1.70}\downarrow$ | $4.66_{\pm 2.49}\downarrow$ | $3.84_{\pm 1.91}\downarrow$ | $7.83_{\pm 4.23}\downarrow$ | $5.64_{\pm 1.40}\downarrow$ |
| FlexMatch[NeurIPS'21] | $21.95_{\pm 0.57}$ | $11.95_{\pm 0.30}$ | $5.42_{\pm 2.95}$ | $4.50_{\pm 2.60}$ | $9.72_{\pm 1.63}$ | $5.85_{\pm 0.98}$ |
| FlexMatch + Ours | $19.74_{\pm 0.50}\downarrow$ | $11.61_{\pm 0.29}\downarrow$ | $\mathbf{2.26}_{\pm 1.13}\downarrow$ | $3.21_{\pm 1.79}\downarrow$ | $8.89_{\pm 3.39}\downarrow$ | $4.97_{\pm 1.19}\downarrow$ |
| FreeMatch[ICLR'23] | $18.27_{\pm 1.60}$ | $11.56_{\pm 0.44}$ | $3.49_{\pm 1.39}$ | $3.22_{\pm 0.55}$ | $10.49_{\pm 1.87}$ | $5.24_{\pm 0.90}$ |
| FreeMatch + Ours | $\underline{14.86}_{\pm 1.22}\downarrow$ | $\underline{10.35}_{\pm 0.68}\downarrow$ | $2.82_{\pm 0.81}\downarrow$ | $\mathbf{2.63}_{\pm 0.70}\downarrow$ | $\underline{3.74}_{\pm 0.90}\downarrow$ | $\underline{\mathbf{3.50}}_{\pm 1.02}\downarrow$ |

We would like to stress that our goal is not to propose a novel state-of-the-art SSL approach, but to shed light about an important issue in prevalent pseudo-labeling based SSL methods, and present a simple yet efficient solution that can improve their performance. These results demonstrate that integrating the penalty in eq. (8) during training appears as an appealing strategy to improve both accuracy and calibration performance of pseudo-label SSL approaches.

Lastly, we follow (Wang et al., 2022) and employ the Friedman rank (Friedman, 1937; 1940) to fairly compare the performance of different methods across various settings. This metric can be defined as $\text{rank}_F = \frac{1}{m}\sum_{i=1}^{m} \text{rank}_i$, with $m$ being the number of evaluation settings ($m = 12$ in our case, 2 metrics $\times$ 6 datasets), and $\text{rank}_i$ the rank of a method in the $i$-th setting. Hence, the lower the rank obtained, the better the method. These rankings, which are depicted in fig. 4, show that our modified versions provide competitive performances considering accuracy (Error), ECE and both (All), with our FreeMatch and FlexMatch versions presenting the best alternatives across overall methods.

**On the impact on the logits.** We now analyze in more detail the impact of incorporating the constraint during training, particularly on the logit distribution. In fig. 5, we depict the kernel density estimation of the logits distribution, which was negatively affected by SSL methods compared to a fully-supervised baseline (*Observation 3*). From these figures, we can observe two interesting findings that will have a positive effect

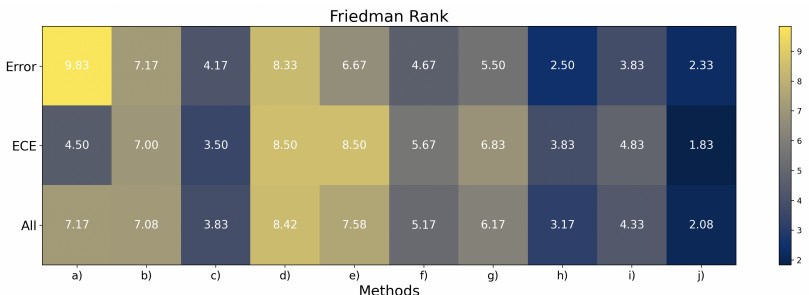

Figure 4: **Friedman Rank** for the methods analyzed in Tables 1 and 2 following (Wang et al., 2022): **a)** MixMatch, **b)** Dash, **c)** AdaMatch, **d)** DeFixMatch, **e)** Fixmatch, **f)** FixMatch + Ours, **g)** FlexMatch, **h)** FlexMatch + Ours, **i)** FreeMatch and **j)** FreeMatch + Ours.

in calibration: *i)* the logit magnitude of the incorrect predicted classes is decreased, and *ii)* the whole logit range also decreases. This means that, even for incorrect predictions, the reduced logit values will lead to less confident predictions, which contrasts with the potentially more confidence scores obtained by SSL methods. We argue that the reduction of the logit magnitudes is a byproduct of the penalty preventing the logit differences to be large. This limits the effect of the min-entropy pushing hard towards the vertex of the simplex, which is minimized with either 0 or 1 softmax predictions.

Thus, following our hypothesis from Observation 3, we can advocate that our strategy provides well-calibrated SSL methods, as both the total range and logit magnitude of incorrect predictions are significantly decreased.

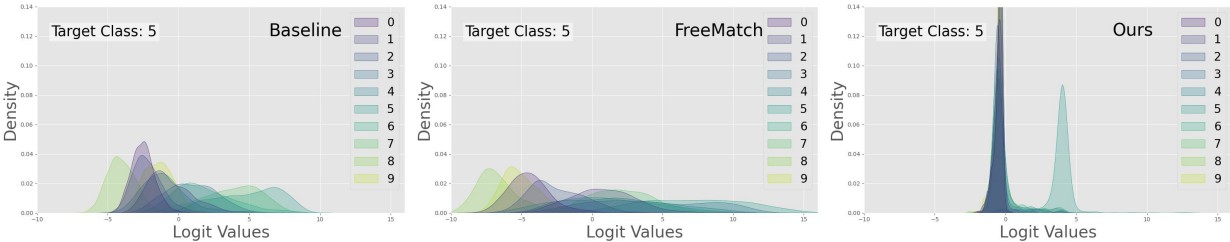

Figure 5: **Impact of the proposed solution in the logits**, which plots the Kernel density estimation of the logits distribution (per-class) for target class 5 for the supervised baseline (*left*), original FreeMatch (*middle*) and our version (*right*).

**Comparison to other calibration methods**. In this section, we compare our approach to relevant calibration approaches in the fully-supervised scenario. In particular, we evaluate the impact of adding label smoothing (LS) (Szegedy et al., 2016) and focal loss (FL) (Lin et al., 2017) to FreeMatch, as it is the most recent studied method. Results from these experiments, depicted in fig. 6, confirm that our approach consistently yields the best discriminative-calibration trade-off across datasets and settings, which can be measured by the largest gap between the accuracy (*bright yellow*) and ECE (*dark yellow*). Furthermore, we compare to BAM (Loh et al., 2023), up to our knowledge the only concurrent work that tackles calibration work on SSL. This comparison (table 3) demonstrates that our simple solution outperforms the recent BAM in terms of error rate and ECE, emerging as a promising choice.

Table 3: Discriminative and calibration performance compared to BAM (Loh et al., 2023).

| Method | CIFAR100 (200) | | CIFAR100 (400) | |
|---|---|---|---|---|
| | Error | ECE | Error | ECE |
| FixMatch + BAM | $28.46_{\pm1.74}$ | $25.04_{\pm1.43}$ | $19.32_{\pm0.94}$ | $16.75_{\pm0.96}$ |
| FixMatch + Ours | $\mathbf{27.57_{\pm1.49}}$ | $\mathbf{21.56_{\pm1.32}}$ | $\mathbf{18.42_{\pm1.65}}$ | $\mathbf{12.12_{\pm1.70}}$ |

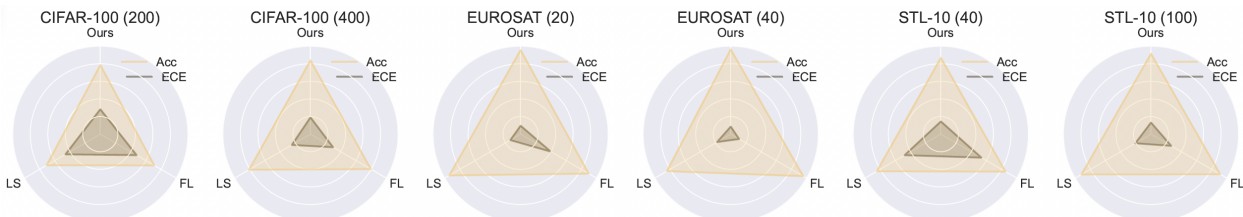

Figure 6: Radar plots for different calibration approaches based on FreeMatch. The range of the radar plot for ECE is changed for better visualization (CIFAR-100: [1, 50], EuroSAT: [1, 20], STL-10: [1, 20])

**In which samples to impose the penalty?** A natural question that arises from our analysis is whether constraining the logit distances should also be applied to the samples in $\mathcal{D}_{U'}$. We argue that, indeed, it is only beneficial to the samples in $\mathcal{D}_{U''}$. While we formally exposed that the third term in eq. (8) minimizes the min-entropy, the second term has some sort of corrective effect, where pseudo-labels from weak augmentations correct the predictions obtained with the strong augmented versions. As labels are different, enforcing a penalty on the logit distances might indeed have a counterproductive effect. More concretely, trying to satisfy the constraint in samples from $\mathcal{D}_{U'}$, where hard predictions are from different classes, may actually impede the network to learn semantically meaningful features that can bring discriminative capabilities to the model. To support these arguments empirically, we report in table 4 the performance of the three SSL methods - FixMatch, FlexMatch and FreeMatch when the constraint is enforced across different scenarios. We can clearly see that, regardless of the configuration or model used, including the penalty on the samples from $\mathcal{D}_{U'}$ has a detrimental effect, with discriminative and calibration results suffering a substantial degradation.

Table 4: **Impact of the penalty term** in the different unlabeled subsets, $\mathcal{D}_{\mathcal{U}'}$ and $\mathcal{D}_{\mathcal{U}''}$ for the three relevant pseudo-labeling SSL methods studied in this work. The proposed approach is shadowed in gray and best result in bold.

| | $\mathcal{D}_{U'}$ | $\mathcal{D}_{U''}$ | CIFAR-100 (200) | | CIFAR-100 (400) | |
| --- | --- | --- | --- | --- | --- | --- |
| | | | Error | ECE | Error | ECE |
| FixMatch | | ✓ | $27.77_{\pm 1.49}$ | $31.28_{\pm 1.58}$ | $15.76_{\pm 0.39}$ | $19.73_{\pm 0.56}$ |
| | | ✓ | $21.56_{\pm 1.32}$ | $27.57_{\pm 1.49}$ | $13.49_{\pm 0.80}$ | $\mathbf{18.57}_{\pm 0.77}$ |
| | ✓ | ✓ | $\mathbf{19.30}_{\pm 2.52}$ | $\mathbf{25.72}_{\pm 2.36}$ | $\mathbf{13.00}_{\pm 1.79}$ | $19.20_{\pm 1.63}$ |
| FlexMatch | | ✓ | $21.95_{\pm 0.57}$ | $28.27_{\pm 0.59}$ | $11.95_{\pm 0.30}$ | $\mathbf{17.61}_{\pm 0.51}$ |
| | | ✓ | $\mathbf{19.74}_{\pm 0.50}$ | $\mathbf{26.49}_{\pm 0.52}$ | $\mathbf{11.61}_{\pm 0.29}$ | $18.15_{\pm 0.47}$ |
| | ✓ | ✓ | $20.98_{\pm 3.05}$ | $27.57_{\pm 3.37}$ | $13.41_{\pm 0.51}$ | $19.62_{\pm 0.44}$ |
| FreeMatch | | ✓ | $23.92_{\pm 2.02}$ | $18.27_{\pm 1.95}$ | $16.18_{\pm 0.38}$ | $11.56_{\pm 0.53}$ |
| | | ✓ | $\mathbf{21.36}_{\pm 1.62}$ | $\mathbf{14.86}_{\pm 1.48}$ | $\mathbf{16.09}_{\pm 0.80}$ | $\mathbf{10.35}_{\pm 0.83}$ |
| | ✓ | ✓ | $27.44_{\pm 1.08}$ | $20.86_{\pm 1.48}$ | $18.82_{\pm 0.67}$ | $12.39_{\pm 0.78}$ |

**Can we add our calibration strategy to other methods?**. We have evaluated the impact of incorporating the proposed term into relevant SSL approaches that resort to hard pseudo-labels during training. Indeed, our motivation stems from the observations that under this scenario (i.e., the standard pseudo-labeling process in SSL), there is a hidden min-entropy objective that dominates the training. While we have demonstrated that our approach effectively improves both the discriminative and calibration performance of these models, we further assess its impact on approaches that use soft pseudo-labels, i.e., pseudo-labels on weak augmentations are not converted to one-hot encoded vectors. For these approaches, the reasoning in eq. (5) and eq. (6) does not entirely hold in these cases. More concretely, the recent work SimMatch (Zheng et al., 2022) uses this strategy[3], where the direct softmax prediction over the weak augmented samples is used as supervisory signal for its strong augmented counterpart. On the other hand, for SoftMatch (Chen et al., 2023), the weighted thresholding mask prevents overly confident predictions, distinguishing these methods from those relying entirely on hard pseudo-labels. The results from this experiment are reported in Table 5, which show that adding the proposed penalty has a positive effect favouring better calibrated models even

---

[3]Note that most pseudo-labeling SSL approaches use a hard pseudo-label strategy, where the softmax predictions of weak augmentations are transformed into one-hot encoded pseudo-labels.

in methods that use soft pseudo-labels, particularly as the number of labeled samples increases. Note that, as our main findings, observations, and motivations, do not entirely apply for soft pseudo-labels approaches, such as SimMatch and SoftMatch, we do not make any claim regarding the benefits of our strategy in these approaches. In contrast, we simply show empirically that even in these cases, our method can still bring performance benefits.

Table 5: Comparison to related approaches, SimMatch and SoftMatch, which use soft pseudo-labels, instead of hard pseudo-labels (as FixMatch, FlexMatch and FreeMatch).

| | Error rate (%) | | | ECE | | |
|---|---|---|---|---|---|---|
| | CIFAR-100 | | | | | |
| # Labeled samples | 200 | 400 | 1000 | 200 | 400 | 1000 |
| SimMatch | $\mathbf{22.49}_{\pm 0.26}$ | $19.45_{\pm 0.08}$ | $15.46_{\pm 0.48}$ | $\mathbf{3.17}_{\pm 0.28}$ | $6.54_{\pm 1.92}$ | $9.60_{\pm 0.72}$ |
| SimMatch + Ours | $23.00_{\pm 0.07}\uparrow$ | $\mathbf{19.10}_{\pm 0.31}\downarrow$ | $\mathbf{15.36}_{\pm 0.41}\downarrow$ | $3.31_{\pm 0.59}\uparrow$ | $\mathbf{5.55}_{\pm 0.13}\downarrow$ | $\mathbf{9.18}_{\pm 0.17}\downarrow$ |
| SoftMatch | $22.69_{\pm 1.74}$ | $16.49_{\pm 0.83}$ | $14.49_{\pm 0.35}$ | $\mathbf{17.64}_{\pm 1.27}$ | $12.09_{\pm 0.82}$ | $10.63_{\pm 0.33}$ |
| SoftMatch + Ours | $21.72_{\pm 0.31}\downarrow$ | $\mathbf{16.82}_{\pm 0.74}\uparrow$ | $\mathbf{14.53}_{\pm 0.36}\uparrow$ | $15.80_{\pm 0.32}\downarrow$ | $\mathbf{11.70}_{\pm 0.86}\downarrow$ | $\mathbf{10.60}_{\pm 0.46}\downarrow$ |

**Long-tailed experiments**. In table 6 we report error rate and ECE on CIFAR100-LT to asses the effect of the proposed calibration when the class population is imbalanced. Across all cases both the method accuracy and the ECE improve consistently. It is noteworthy to mention that for all these experiments we use the same margin as in CIFAR-100, without any adaptation to the specific setting.

Table 6: Qualitative performance on long-tailed classification (CIFAR-100-LT).

| Method | $\gamma_l = 10, \gamma_u = -10$ | | $\gamma_l = 10, \gamma_u = 10$ | | $\gamma_l = 15, \gamma_u = 15$ | |
|---|---|---|---|---|---|---|
| | Error | ECE | Error | ECE | Error | ECE |
| FixMatch | $15.05_{\pm 0.13}$ | $8.58_{\pm 0.82}$ | $15.01_{\pm 0.18}$ | $10.51_{\pm 1.82}$ | $16.45_{\pm 0.11}$ | $10.22_{\pm 2.01}$ |
| FixMatch + Ours | $\mathbf{14.73}_{\pm 0.12}$ | $\mathbf{7.22}_{\pm 1.20}$ | $\mathbf{14.37}_{\pm 0.21}$ | $\mathbf{9.42}_{\pm 0.29}$ | $\mathbf{15.81}_{\pm 0.31}$ | $\mathbf{8.44}_{\pm 0.44}$ |
| FlexMatch | $14.79_{\pm 0.28}$ | $7.70_{\pm 1.03}$ | $14.98_{\pm 0.21}$ | $9.49_{\pm 0.47}$ | $16.19_{\pm 0.32}$ | $9.25_{\pm 0.68}$ |
| FlexMatch + Ours | $\mathbf{14.72}_{\pm 0.23}$ | $\mathbf{7.33}_{\pm 1.49}$ | $\mathbf{14.6}_{\pm 0.05}$ | $\mathbf{7.44}_{\pm 0.40}$ | $\mathbf{16.08}_{\pm 0.49}$ | $\mathbf{8.85}_{\pm 0.47}$ |
| FreeMatch | $14.93_{\pm 0.20}$ | $8.52_{\pm 0.95}$ | $15.04_{\pm 0.09}$ | $9.99_{\pm 0.52}$ | $16.26_{\pm 0.35}$ | $8.68_{\pm 0.84}$ |
| FreeMatch + Ours | $\mathbf{14.67}_{\pm 0.28}$ | $\mathbf{6.74}_{\pm 0.98}$ | $\mathbf{14.77}_{\pm 0.15}$ | $\mathbf{9.05}_{\pm 0.84}$ | $\mathbf{15.87}_{\pm 0.69}$ | $\mathbf{8.53}_{\pm 0.60}$ |

**Additional experiments** with detailed analysis across methods, comparison to additional approaches and additional settings are reported in Appendix.

**Scalability and Computational Costs.** The proposed approach boils down to a simple penalty term, which can be straightforwardly integrated on existing methods by modifying the loss function, whose computational cost is negligible. Indeed, our learning approach does not involve architectural changes, avoiding any increase in model complexity. Thus, the strategy proposed in this work further improves the discriminative and calibration performance of existing SSL methods without incurring additional overheads.

## 6 Conclusion

In this work we have raised awareness of the miscalibration problem induced by pseudo-label training, which is one of the most popular approaches for SSL. We demonstrated that the unsupervised loss used by pseudo-label methods is dominated by the min-entropy term, a lower bound of the Shannon entropy, and identified it as a potential source of miscalibration. We then proposed a simple solution based on enforcing a fixed margin constraint between the winner class and its contenders. Our solution on popular SSL datasets and methods yields consistent calibration improvements, whereas we also found consistent gains in terms of predictive accuracy, typically outperforming the state-of-the-art for SSL.

## Acknowledgments

This work was funded by the Natural Sciences and Engineering Research Council of Canada (NSERC). We also thank Calcul Quebec and Compute Canada.

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

# A  Additional discriminative results

Table 7: Comparison of error rate (%) for different SSL methods across various labeled settings and datasets (CIFAR-100, EuroSAT, and STL-10).† indicates that the results are reported from (Wang et al., 2022). Best results in bold, whereas second-best are underlined.

| Dataset | CIFAR-100 | | EuroSAT | | STL-10 | |
|---|---|---|---|---|---|---|
| # Labeled samples | 200 | 400 | 20 | 40 | 40 | 100 |
| $\pi$-Model $_{\text{NeurIPS'15}}$ † | $36.06_{\pm 0.15}$ | $26.52_{\pm 0.41}$ | $21.82_{\pm 1.22}$ | $12.09_{\pm 2.27}$ | $42.76_{\pm 15.94}$ | $19.85_{\pm 13.02}$ |
| Mean-Teacher $_{\text{NeurIPS'17}}$ | $35.47_{\pm 0.40}$ | $26.03_{\pm 0.30}$ | $26.83_{\pm 1.46}$ | $15.85_{\pm 1.66}$ | $18.67_{\pm 2.66}$ | $24.19_{\pm 10.15}$ |
| VAT $_{\text{TPAMI'18}}$ † | $31.49_{\pm 1.33}$ | $21.34_{\pm 0.50}$ | $18.45_{\pm 1.47}$ | $26.16_{\pm 0.96}$ | $10.09_{\pm 0.94}$ | $10.69_{\pm 0.51}$ |
| MixMatch $_{\text{NeurIPS'19}}$ | $37.68_{\pm 2.66}$ | $26.84_{\pm 1.06}$ | $28.77_{\pm 10.40}$ | $14.88_{\pm 2.07}$ | $25.19_{\pm 2.05}$ | $11.37_{\pm 1.49}$ |
| ReMixMatch $_{\text{ICLR'20}}$† | $22.21_{\pm 2.21}$ | $16.86_{\pm 0.57}$ | $5.05_{\pm 1.05}$ | $5.07_{\pm 0.56}$ | $13.08_{\pm 3.34}$ | $\mathbf{7.21}_{\pm 0.39}$ |
| UDA $_{\text{NeurIPS'20}}$ | $28.80_{\pm 0.61}$ | $19.00_{\pm 0.79}$ | $9.83_{\pm 2.15}$ | $6.22_{\pm 1.36}$ | $15.58_{\pm 3.16}$ | $7.65_{\pm 1.11}$ |
| CRMatch $_{\text{GCPR'21}}$† | $25.70_{\pm 1.75}$ | $18.03_{\pm 0.20}$ | $13.24_{\pm 1.69}$ | $8.35_{\pm 1.71}$ | $\mathbf{10.17}_{\pm 0.00}$ | − |
| CoMatch $_{\text{ICCV'21}}$† | $35.08_{\pm 0.69}$ | $25.35_{\pm 0.50}$ | $5.75_{\pm 0.43}$ | $4.81_{\pm 1.05}$ | $15.12_{\pm 1.88}$ | $9.56_{\pm 1.35}$ |
| Dash $_{\text{ICML'21}}$ | $28.51_{\pm 2.91}$ | $19.54_{\pm 1.20}$ | $10.05_{\pm 8.15}$ | $6.83_{\pm 3.24}$ | $18.30_{\pm 4.58}$ | $8.74_{\pm 2.13}$ |
| AdaMatch $_{\text{ICLR'22}}$ | $\mathbf{19.26}_{\pm 1.83}$ | $17.13_{\pm 0.92}$ | $12.01_{\pm 4.16}$ | $6.07_{\pm 2.26}$ | $13.31_{\pm 3.75}$ | $8.14_{\pm 1.48}$ |
| SimMatch $_{\text{CVPR'22}}$ † | $23.78_{\pm 1.08}$ | $17.06_{\pm 0.78}$ | $7.66_{\pm 0.60}$ | $5.27_{\pm 0.89}$ | $\underline{11.77}_{\pm 3.20}$ | $7.55_{\pm 1.86}$ |
| SoftMatch $_{\text{ICLR'23}}$ † | $22.67_{\pm 1.32}$ | $16.84_{\pm 0.66}$ | $5.75_{\pm 0.62}$ | $5.90_{\pm 1.42}$ | $13.55_{\pm 3.16}$ | $7.84_{\pm 1.72}$ |
| DeFixMatch $_{\text{ICLR'23}}$ | $30.44_{\pm 0.82}$ | $20.93_{\pm 1.42}$ | $14.27_{\pm 9.05}$ | $5.42_{\pm 2.69}$ | $25.36_{\pm 4.40}$ | $10.97_{\pm 1.75}$ |
| FixMatch $_{\text{NeurIPS'20}}$ | $31.28_{\pm 1.58}$ | $19.73_{\pm 0.56}$ | $11.88_{\pm 6.32}$ | $6.64_{\pm 5.03}$ | $16.13_{\pm 2.36}$ | $8.06_{\pm 2.15}$ |
| FixMatch + Ours | $27.57_{\pm 1.49}$ | $18.57_{\pm 0.77}$ | $7.19_{\pm 4.83}$ | $5.02_{\pm 2.24}$ | $17.55_{\pm 4.00}$ | $7.96_{\pm 1.64}$ |
| FlexMatch $_{\text{NeurIPS'21}}$ | $28.27_{\pm 0.59}$ | $17.61_{\pm 0.51}$ | $7.89_{\pm 3.06}$ | $7.13_{\pm 1.23}$ | $13.34_{\pm 1.63}$ | $8.35_{\pm 1.24}$ |
| FlexMatch + Ours | $26.49_{\pm 0.52}$ | $18.15_{\pm 0.47}$ | $\mathbf{3.69}_{\pm 0.81}$ | $5.00_{\pm 0.98}$ | $12.87_{\pm 4.32}$ | $\underline{7.53}_{\pm 1.32}$ |
| FreeMatch $_{\text{ICLR'23}}$ | $23.92_{\pm 2.02}$ | $\underline{16.18}_{\pm 0.38}$ | $4.74_{\pm 1.77}$ | $4.48_{\pm 0.73}$ | $14.88_{\pm 0.72}$ | $8.83_{\pm 0.14}$ |
| FreeMatch + Ours | $\underline{21.36}_{\pm 1.62}$ | $\mathbf{16.09}_{\pm 0.80}$ | $\underline{4.30}_{\pm 1.46}$ | $\mathbf{3.50}_{\pm 0.70}$ | $13.18_{\pm 1.61}$ | $8.57_{\pm 1.05}$ |

In the main paper, we included several relevant SSL methods to compare both error and calibration performance. We did not include a considerably large set of approaches because we need to run three times each method in each setting (so that each evaluated method means 3×6=18 runs, with an average of 10 hours per run). Furthermore, we would like to stress again that our goal in this work is not to provide a novel state-of-the-art SSL method, but investigate the miscalibration issue of pseudo-labeling SSL, identify the source of the problem, and provide a solution that can enhance the calibration of this family of methods. Having said this, the discriminative performance of additional methods is provided in (Wang et al., 2022), which is used to complement the discriminative comparison in Table 1, whose results can be found in Table 7 of this appendix.

# B Dataset details

We provide detailed insights into the datasets utilized in our experiments, whose configuration is strongly inspired by the recent USB (A Unified Semi-supervised Learning Benchmark for Classification) benchmark (Wang et al., 2022). For **CIFAR-100**, a renowned benchmark for fine-grained image classification, we considered two label settings: 2 labeled samples and 4 labeled samples per class for each of the 100 classes,resulting in a total of 50,000 training samples and 10,000 samples for testing. Each image in CIFAR-100 is sized at $32{\times}32$ pixels. **STL-10**, known for its limited sample size and extensive unlabeled data, offers a unique challenge. We employed two label settings as well: 4 labeled samples and 10 labeled samples per class for all 10 classes, and an additional 100,000 unlabeled samples for training, along with 8,000 samples for testing. Each STL-10 image measures $96{\times}96$ pixels. Lastly, **EuroSAT**, based on Sentinel-2 satellite images, features two label settings: 2 labeled samples per class and 4 samples per class for 10 classes. With a total of 16,200 training samples, including labeled and unlabeled images and 5,400 testing samples, EuroSAT images are sized at $64{\times}64$. While literature includes SVHN and CIFAR-10 datasets, we do not use them for evaluation, as state-of-the-art SSL methods already achieve a performance comparable to that of fully supervised training on these datasets. The selected datasets offer diverse challenges and settings, allowing for a comprehensive evaluation of semi-supervised learning methods across various domains.

For the **long-tailed CIFAR-100** experiments, we investigate two subsets of the dataset, with $N1$ representing the number of samples in the minority class and $M$ denoting the number of samples in the majority class. The imbalance ratio is controlled by $\gamma_l$ and $\gamma_u$, where both parameters are set to -10, 10 or 15, ensuring a consistent level of label imbalance across categories. In the first setting, we set the imbalance ratio of labeled samples $\gamma_l$ to 10 and the imbalance ratio of unlabeled samples $\gamma_u$ to 10, with the number of labeled samples (N1) set to 150 and the number of unlabeled samples (M) set to 300. In the second setting, we maintain the same label imbalance ratio ($\gamma_l = 10$) and number of labeled samples (N1 = 150), but we set the unlabeled sample imbalance ratio $\gamma_u$ to -10. Finally, in the third setting, we increase the $\gamma_l$ to 15 while keeping the $\gamma_u$ at 15, with N1 and M set to 150 and 300, respectively. This configuration introduces a higher degree of label imbalance, with the minority class having 15 times fewer samples than the majority class. We follow the same setup as authors in (Wang et al., 2022) for the classic setting using WideResnet. These settings allow us to comprehensively evaluate the performance of machine learning models under varying degrees of label imbalance and class distribution scenarios within the CIFAR100-LT dataset.

# C Implementation details

*Method-dependent hyperparemeters.* In the context of pseudo-labeling methods, while FixMatch and FlexMatch rely on a single threshold hyperparameter for pseudo-label selection, FreeMatch introduces additional hyperparameters such as a pre-defined threshold $\tau$, unlabeled batch ratio $\mu$, unsupervised loss weight $w_u$, fairness loss weight $w_f$, and EMA decay $\lambda$. To avoid unfair comparisons across methods, we use the default values for all these hyperparameters, which are reported in their respective papers.

*Specific-case: BAM.* In our comparison with BAM (Loh et al., 2023), we encountered difficulties in hyperparameter tuning. BAM employs the quantile $Q$ over the batch to determine the threshold for pseudo-label selection, which serves as a key hyperparameter. In the cited paper, $Q = 0.75$ for the CIFAR-100 benchmark and $Q = 0.95$ for the CIFAR-10 benchmark were chosen. Additionally, BAM incorporates a separate Adam optimizer for the Bayesian Neural Network (BNN) layer, with a fixed learning rate of 0.01. Notably, for the sharpening temperature parameter in BaM-UDA, BAM opts for $t = 0.9$ as opposed to $t = 0.4$ utilized in UDA. These hyperparameter selections are specific to both the dataset and the method employed, thus presenting a challenge when evaluating the method across different datasets. For our experiments, we chose to focus on CIFAR-100 due to the inherent complexity involved in conducting extensive hyperparameter searches, not only concerning the dataset or method, but also regarding the optimizer and learning rate for the introduced BNN layer.

*Margin Selection with Limited Fine-tuning:* We stress that in the majority of cases, we did not extensively fine-tune the margin hyperparameter. Across various experiments, settings and datasets, the chosen margin values remained consistent without significant adjustments. Specifically, for the CIFAR-100 and EuroSAT

datasets, the margins were uniformly set without requiring further refinement, with values of 10 and 8 across methods, respectively. Given its complexity, we needed to perform a hyperparameter search in a few cases in the STL-10 dataset. In particular, most settings employed a margin equal to 10, similar to the CIFAR-100 dataset. Nevertheless, we observe that, particularly for FreeMatch, this margin did not yield the best performance across the two settings of labeled data (i.e., 40 and 100 labeled samples). After fine-tuning the margin value, we found that for STL-10 (40) and STL-10 (100) FreeMatch worked best with a margin of 4 and 6, respectively. Furthermore, for all the remaining experiments with CIFAR-100, as well as with long-tailed CIFAR-100, we kept the margin fixed ($m = 10$) across experiments and methods.

*Training hyperparemeters.* Regarding algorithm-independent hyperparameters, we adhered to the settings outlined in (Wang et al., 2022). Specifically, the learning rate was set to $5 \times 10^{-4}$ for CIFAR-100, $10^{-4}$ for STL-10, and $5 \times 10^{-5}$ for EuroSAT. During training, the batch size was fixed at 8, while for evaluation, it was set to 16. Additionally, the layer decay rate varied across datasets: 0.5 for CIFAR-100, 0.95 for STL-10, and 1.0 for EuroSAT. Weak augmentation techniques employed included random crop and random horizontal flip, while strong augmentation utilized RandAugment (Cubuk et al., 2020). The cosine annealing scheduler was utilized with a total of 204,800 steps and a warm-up period of 5,120 steps. Both labeled and unlabeled batch sizes were set to 16. Furthermore, we follow (Wang et al., 2022; Zhang et al., 2021) to report the best number of all checkpoints to avoid unfair comparisons caused by different convergence speeds.

## D Model choices

In this work we have selected three relevant SSL approaches based on hard pseudo-labels, as our findings are closely related to these approaches. More concretely, we demonstrate empirically our observations in FixMatch (Sohn et al., 2020), FlexMatch (Zhang et al., 2021) and FreeMatch (Wang et al., 2023), as they are popular SSL methods, some of them published recently (i.e., FreeMatch has been published in 2023). Thus, for the Table 1 and Table 2, we have shown the effect of adding the proposed term in all the three approaches. Similarly, the impact of our term over the three methods is evaluated in Table 6 (i.e., CIFAR-100-LT), as we consider that long-tailed classification represents an important task, and assessing the performance of the three approaches will undoubtedly strengthen the message that our solution can improve both classification and calibration performance across a general family of SSL approaches. Regarding the ablation studies to evaluate the impact of several choices, we have used FreeMatch, the most recent approach among the three, in all the ablations (Fig 6 and Table 4), except when comparing to the concurrent work in BAM (Loh et al., 2023). The reason is that authors in (Loh et al., 2023) only consider FixMatch in their experiments, despite BAM being a work published in 2023. Thus, the code[4] is all constrained to the use of FixMatch and CIFAR-100, making it difficult to evaluate with other SSL approaches and datasets. In fact, we attempted to integrate FreeMatch on the BAM framework, but the results obtained were suboptimal. We believe that, the sensitivity to hyperparameters (e.g., just the choice of the backbone for the Bayesian layer in BAM requires different learning rates) could be the main cause for the low performance of BAM in SSL approaches other than FixMatch. Therefore, in order to avoid an unfair comparison (due to the suboptimal performance of BAM in other SSL methods) we decided to compare to it based only on FixMatch, following their original work.

## E Numerical values for results in radar plots

For the radar plots shown in the main paper, we present here (Table 8) the individual quantitative results, for a more detailed comparison across methods. As a reminder, FreeMatch (Wang et al., 2023) is used as a baseline strategy, and the different calibration strategies (i.e., Focal Loss (Lin et al., 2017), Label Smoothing (Szegedy et al., 2016), and ours) are applied over the $\mathcal{D}_{U''}$ subset.

---

[4]https://github.com/clott3/BaM-SSL

Table 8: Numerical values of radar plots depicted in Figure 6 of the main paper.

| Method | Setting | Error | ECE | CECE | AECE |
|---|---|---|---|---|---|
| Baseline | | $23.92_{\pm2.02}$ | $18.27_{\pm1.95}$ | $0.41_{\pm0.04}$ | $18.27_{\pm1.95}$ |
| Focal Loss | CIFAR 100 (200) | $28.81_{\pm0.75}$ | $24.42_{\pm0.85}$ | $0.53\pm_{0.01}$ | $24.42_{\pm0.85}$ |
| Label Smoothing | | $29.14_{\pm2.46}$ | $23.55_{\pm2.33}$ | $0.52_{\pm0.04}$ | $23.54_{\pm2.34}$ |
| **Ours** | | $\mathbf{21.36}_{\pm1.62}$ | $\mathbf{14.86}_{\pm1.48}$ | $\mathbf{0.35}_{\pm0.03}$ | $\mathbf{14.81}_{\pm1.48}$ |
| Baseline | | $16.18_{\pm0.38}$ | $11.56_{\pm0.53}$ | $0.27_{\pm0.01}$ | $11.54_{\pm0.52}$ |
| Focal Loss | CIFAR 100 (400) | $20.35_{\pm0.35}$ | $15.63_{\pm0.54}$ | $0.35_{\pm0.008}$ | $15.60_{\pm0.54}$ |
| Label Smoothing | | $19.01_{\pm1.90}$ | $13.03_{\pm1.39}$ | $0.31_{\pm0.02}$ | $13.02_{\pm1.39}$ |
| **Ours** | | $\mathbf{16.09}_{\pm0.80}$ | $\mathbf{10.35}_{\pm0.83}$ | $\mathbf{0.26}_{\pm0.01}$ | $\mathbf{10.33}_{\pm0.83}$ |
| Baseline | | $4.74_{\pm1.82}$ | $3.50_{\pm1.70}$ | $0.82_{\pm0.32}$ | $3.38_{\pm1.82}$ |
| Focal Loss | EuroSAT (20) | $10.25_{\pm8.48}$ | $8.54_{\pm8.29}$ | $1.88_{\pm1.70}$ | $8.52_{\pm8.31}$ |
| Label Smoothing | | $5.94_{\pm4.21}$ | $3.57_{\pm2.83}$ | $0.95_{\pm0.67}$ | $3.45_{\pm2.81}$ |
| **Ours** | | $\mathbf{4.30}_{\pm1.46}$ | $\mathbf{2.82}_{\pm1.00}$ | $\mathbf{0.70}_{\pm0.23}$ | $\mathbf{2.78}_{\pm0.98}$ |
| Baseline | | $4.48_{\pm0.73}$ | $3.22_{\pm0.55}$ | $0.74_{\pm0.11}$ | $3.15_{\pm0.61}$ |
| Focal Loss | EuroSAT (40) | $4.37_{\pm0.30}$ | $3.10_{\pm0.10}$ | $0.74\pm_{0.04}$ | $3.01_{\pm0.08}$ |
| Label Smoothing | | $15.84_{\pm3.57}$ | $4.43_{\pm3.38}$ | $1.02_{\pm0.70}$ | $4.37_{\pm3.31}$ |
| **Ours** | | $\mathbf{3.50}_{\pm0.70}$ | $\mathbf{2.63}_{\pm0.70}$ | $\mathbf{0.60}_{\pm0.12}$ | $\mathbf{2.58}_{\pm0.64}$ |
| Baseline | | $14.88_{\pm0.72}$ | $10.49_{\pm1.87}$ | $2.45_{\pm0.40}$ | $9.40_{\pm1.81}$ |
| Focal Loss | STL 10 (40) | $14.73_{\pm2.77}$ | $11.16_{\pm3.13}$ | $2.39\pm0.64$ | $11.15_{\pm3.13}$ |
| Label Smoothing | | $\mathbf{15.59}_{\pm2.74}$ | $10.09_{\pm1.64}$ | $\mathbf{2.20}_{\pm0.28}$ | $10.04_{\pm1.58}$ |
| **Ours** | | $\mathbf{13.18}_{\pm1.61}$ | $\mathbf{3.74}_{\pm5.64}$ | $2.39_{\pm0.68}$ | $\mathbf{8.43}_{\pm4.92}$ |
| Baseline | | $7.62_{\pm1.25}$ | $5.24_{\pm1.09}$ | $1.15_{\pm0.21}$ | $5.19_{\pm1.03}$ |
| Focal Loss | STL 10 (100) | $8.75_{\pm1.59}$ | $6.07_{\pm1.40}$ | $1.35_{\pm0.02}$ | $6.02_{\pm1.35}$ |
| Label Smoothing | | $\mathbf{8.25}_{\pm2.05}$ | $4.94_{\pm1.79}$ | $1.13_{\pm0.35}$ | $4.94_{\pm1.79}$ |
| **Ours** | | $8.57_{\pm1.90}$ | $\mathbf{3.50}_{\pm1.54}$ | $\mathbf{0.94}_{\pm0.27}$ | $\mathbf{4.00}_{\pm1.21}$ |

## F   Comparing our method to temperature scaling

Applying temperature scaling ($p_k = \frac{\exp^{l_k/\tau}}{\sum_{j=1}^{K} \exp^{l_j/\tau}}$) can indeed modify the shape of the softmax distribution, leading to smoother distributions when $\tau > 1$. Nevertheless, compared to the proposed solution, scaling the softmax values during training (before using the softmax probabilities into the cross-entropy loss) presents several drawbacks. **1) No mechanism to control the importance of softened distributions:** Systematically scaling the softmax predictions might be suboptimal in several cases: e.g., reduction of the gradient magnitude, loss of sparsity in the predictions, impaired learning of hard examples, or even magnifying the miscalibration issue, as strong softmax smoothing can lead to meaningless uniform distributions. While applying TS on the softmax values will always introduce a risk of these drawbacks to appear, the use of the proposed constrained term acts as a regularizer of the main objective, whose importance can be controlled by a blending hyperparameter $\lambda$, which introduces more flexibility during training. **2) Difficulty on finding the optimal scaling factor.** Softmax distributions at the beginning of the training (mostly uniform) are very different from those observed typically when the network converges (peaky distributions with one category dominating the softmax vector). Thus, the scaling factor to control the logit distance should also evolve during training (e.g., at the beginning, one might not want to scale the softmax at all, as this could magnify all the problems presented in previous point). For instance, a scaling value of 2 does not result to the same logit distances (or their equivalent softmax predictions) on an almost uniform than on a peaky distribution. On the other hand, the proposed penalty acts on the absolute logit values, enforcing the same constraint across the training (i.e., just penalize those predictions whose logit distances are longer than the given margin), which is easier to adjust. **3) Different behaviour.** Based on the previous point, at the beginning of the training, when the logit distribution is more uniform, the penalty term will not apply to most predictions, allowing the cross-entropy, pseudo cross-entropy and min-entropy terms to guide the training. However, it will start to increasingly act as regularizer as logit distributions become sharper, controlling the logits magnitudes from becoming larger and larger, but allowing their distances to be large enough to have discriminative power. On the other hand, scaling the softmax from the beginning will reduce the softmax values of all the samples systematically, which may impede the network to properly learn. Thus, we can argue that the proposed constrained term offers more control over the logit/softmax distributions than simply scaling the softmax predictions, more flexibility as it offers a mechanism to control the importance of the constraint, and a different behaviour, particularly at the beginning of the training.

To quantify the discussion, we present our results in the Table 9. We also illustrate the logit distribution of the three methods under consideration (a) FreeMatch, (b) FreeMatch + Ours and (c) FreeMatch with logits scaled by a temperature of 2. In these plots, it can be observed that the impact on the logit distribution of our method, and systematically applying temperature scaling to all samples, is different, which empirically supports our hypothesis regarding their different behaviour.

Table 9: Comparison of various metrics across different numbers of labeled samples.

| # Labeled Samples | Temperature $(\tau)$ | ECE | Error |
|---|---|---|---|
| 200 | 2 | $26.63_{\pm 0.93}$ | $29.53_{\pm 1.07}$ |
|  | 5 | $26.58_{\pm 0.31}$ | $27.93_{\pm 0.46}$ |
|  | 10 | $27.46_{\pm 0.67}$ | $28.07_{\pm 0.70}$ |
|  | 20 | $27.55_{\pm 0.92}$ | $27.98_{\pm 1.03}$ |
| 400 | 2 | $17.49_{\pm 1.23}$ | $20.43_{\pm 1.04}$ |
|  | 5 | $18.32_{\pm 0.90}$ | $19.69_{\pm 1.11}$ |
|  | 10 | $18.85_{\pm 0.33}$ | $19.48_{\pm 0.34}$ |
|  | 20 | $18.78_{\pm 0.52}$ | $19.10_{\pm 0.51}$ |

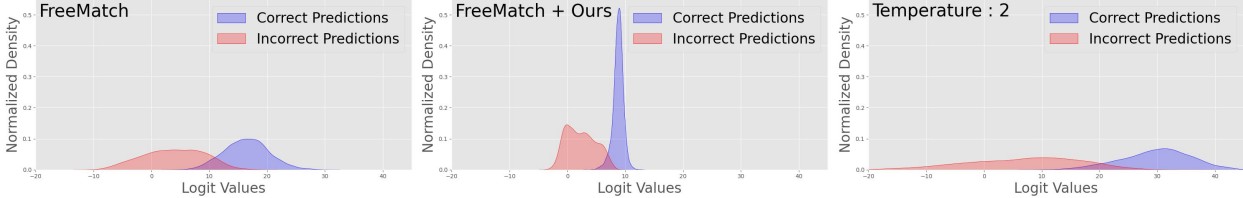

Figure 7: Logit Distribution Plot depicting the distribution of correct predictions vs. incorrect predictions for (a) FreeMatch (b) FreeMatch + Ours and (c) FreeMatch with logits scaled by a temperature of 2 on the CIFAR-100 dataset with 200 labeled samples.

# G  Evaluating our method in a classic setting

As stated in the main paper, we advocate for the use of Visual transformers over standard convolutional neural networks due to their multiple advantages. Nevertheless, to show that our approach is also applicable to convolutional deep models, we evaluate our method against the baseline FreeMatch as well as UPS (Rizve et al., 2021) with WideResNet-28 as the backbone architecture. The results obtained from comparing our method are reported in the Table 10 below. While UPS was initially evaluated on higher amounts of labeled data, semi-supervised learning has evolved to use much lesser data in recent settings, which echoes more realistic scenarios. Thus, we have evaluated UPS under these newer settings, alongside state-of-the-art methods. In particular, for a scenario where 4000 labeled samples are available, and UPS achieves an error rate of 40.77 (See Table 1 in (Rizve et al., 2021)). Thus, it is reasonable to think that its performance is degraded as the amount of labeled samples decreases. Having validated the performance of UPS, we can easily observe that our method, as well as the baseline FreeMatch, significantly outperform UPS in terms of both error rate, and Expected Calibration Error (ECE).

Table 10: Results on FreeMatch with WideResnet - 28 as backbone on CIFAR-100 dataset

|  | Error rate (%) | | ECE | |
|---|---|---|---|---|
|  | CIFAR-100 | | CIFAR-100 | |
| # Labeled samples | 400 | 2500 | 400 | 2500 |
| UPS | 82.67 | 48.85 | 70.43 | 38.55 |
| FreeMatch | 50.17 | 33.35 | 36.05 | 20.63 |
| FreeMatch + Ours | 50.45 | 33.88 | 27.57 | 16.13 |

## H  Detailed comparison between original and our versions

We include additional comparisons between our proposed method and the original SSL approaches.

*Convergence analysis.* First, as depicted in Fig. 8, we assess the impact of adding our approach to FreeMatch in terms of convergence. Notably, our method, denoted as FreeMatch (Ours), demonstrates superior performance in terms of accuracy, outperforming the baseline FreeMatch method within the initial 50,000 iterations. In contrast, the original method reaches convergence around 170,000 iterations, making it considerably slower if one looks at discriminative performance. Additionally, our approach consistently achieves lower values of Expected Calibration Error (ECE) throughout the iterative process, highlighting its potential to enhance the calibration performance. This contrasts with the original FreeMatch, whose predictions seem to be better calibrated at the first iteration, and these are degraded (i.e., ECE increases) with the training. Thus, adding our method exhibits accelerated convergence rates compared to the original approach, reaching competitive performance levels in a fraction of the time.

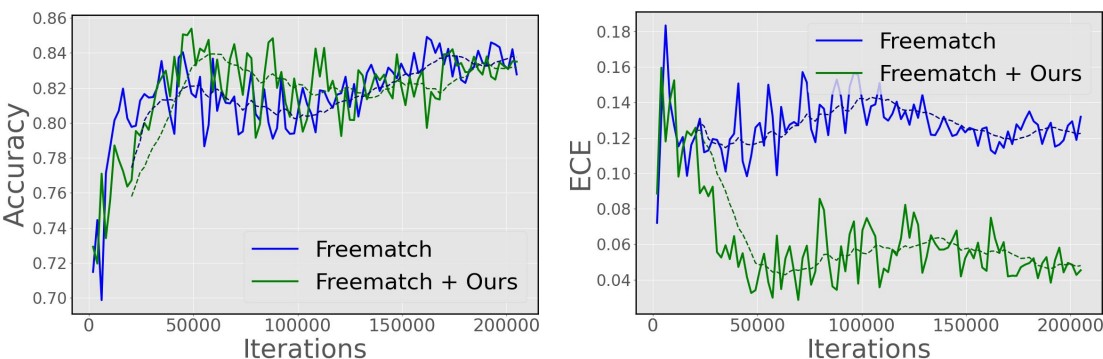

Figure 8: Comparison of convergence of Accuracy (*left*) and ECE (*right*) over iterations for STL10 dataset with 40 labeled samples.

*Logits distribution.* We now provide additional insights into the effects of our approach across the logit distribution. More concretely, in fig. 9 and fig. 10 we present the kernel density estimation of the logits distribution for each class on the STL10 dataset for FreeMatch and FreeMatch plus our method (denoted as Ours). This analysis expands on Observation 3 from the main paper, highlighting the impact of SSL on the calibration of the model across a diverse range of classes. Specifically, while the original FreeMatch provides larger logit ranges, as well as larger logit values even for incorrect classes, adding our approach limits these logit increases across all the classes. As the logit range decreases, the resulting softmax probabilities for the predicted classes will be smaller, alleviating the problem of overconfident predictions. This is particularly important for wrong predictions, as we expect a well-calibrated model to not be overconfident when incorrect classes are predicted.

*Reliability plots.* Last, we depict several reliability plots for both the semi-supervised learning (SSL) methods that were discussed in the main paper and our modified version, to highlight the calibration improvements brought by our approach (fig. 11 and fig. 12).

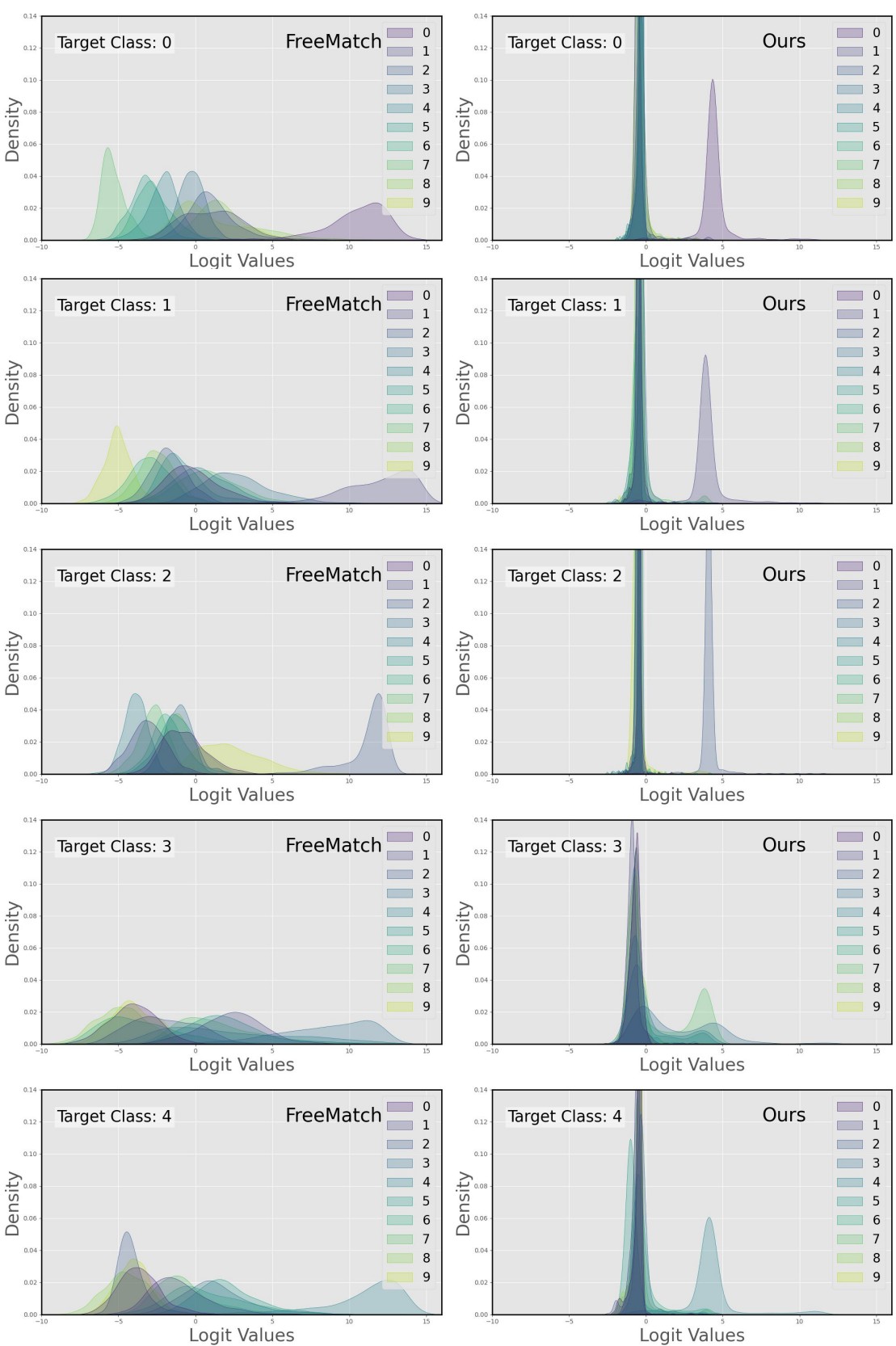

Figure 9: Comparison of convergence of Accuracy (*left*) and ECE (*right*) over iterations for STL10 dataset with 40 labeled samples.

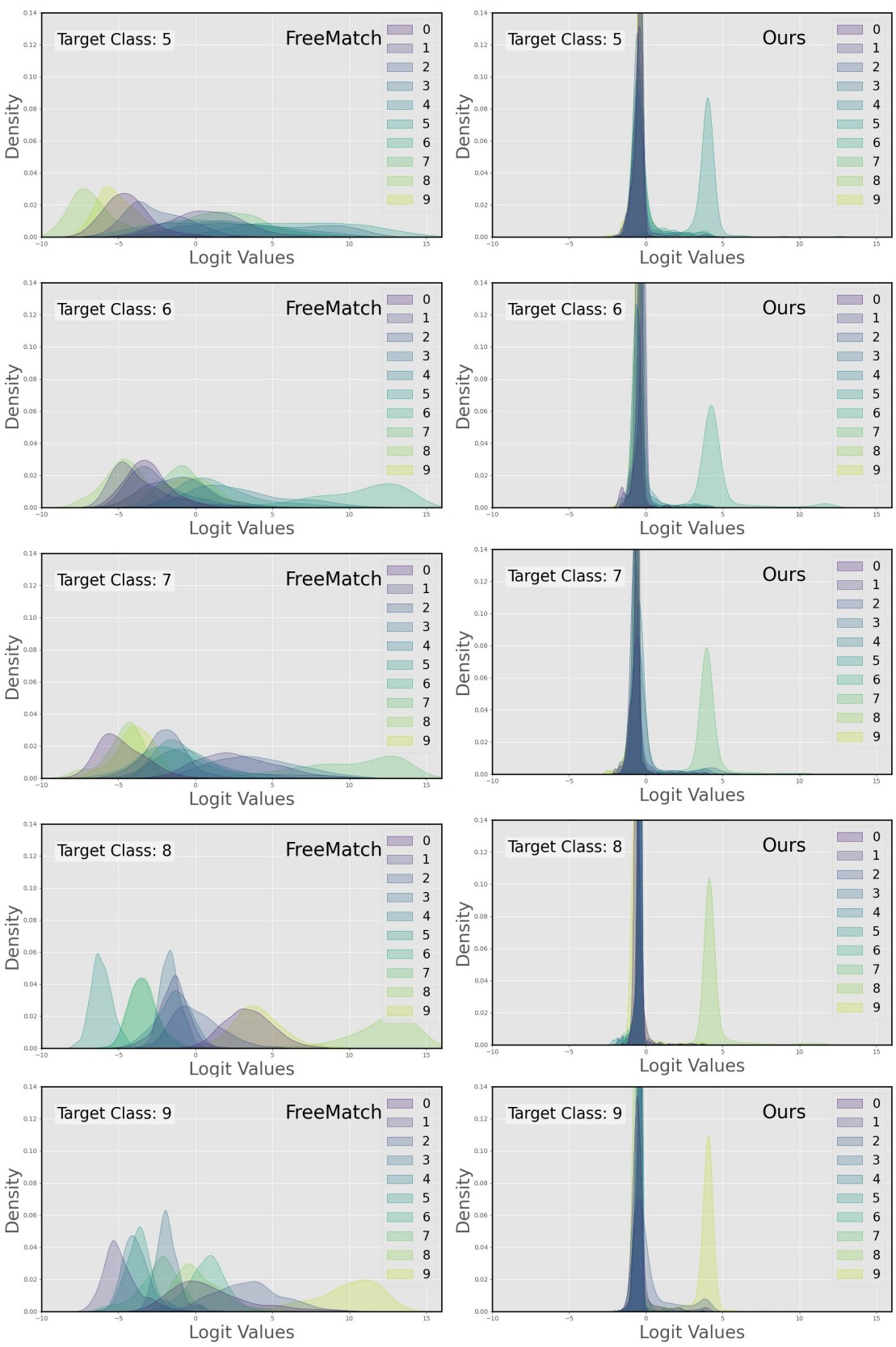

Figure 10: Comparison of convergence of Accuracy (*left*) and ECE (*right*) over iterations for STL10 dataset with 40 labeled samples.

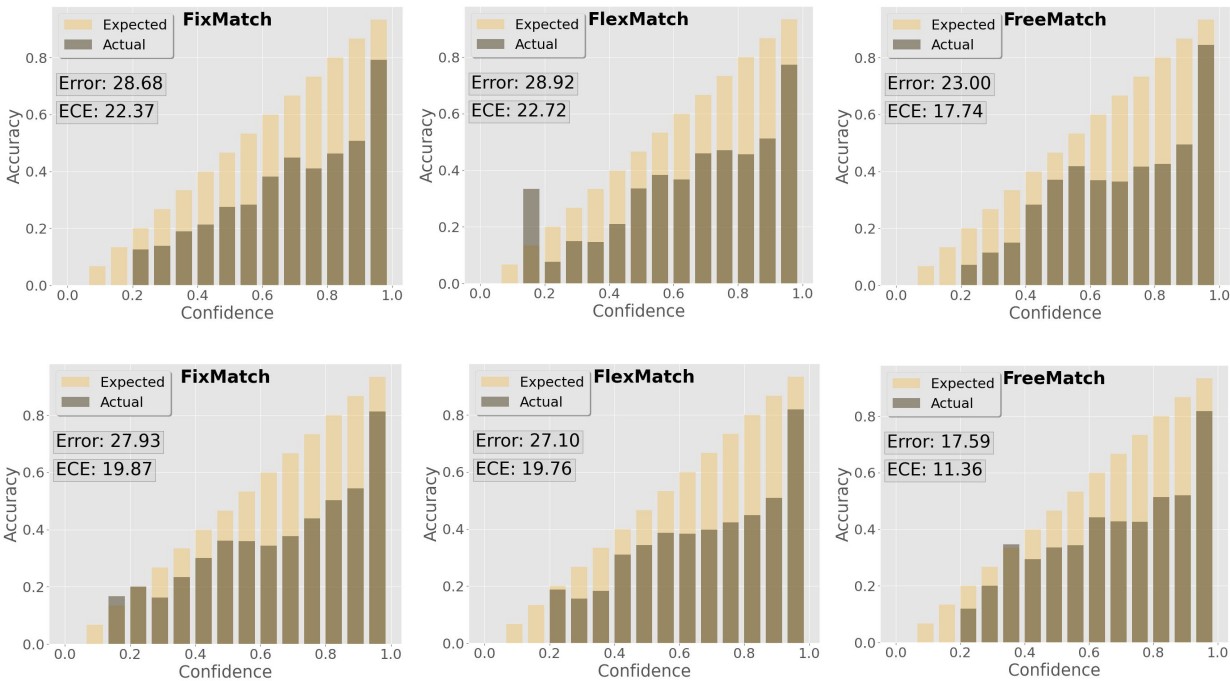

Figure 11: Reliability plots on CIFAR-100(200) for the original (*top*) and our (*bottom*) versions of the three pseudo-label SSL methods selected.

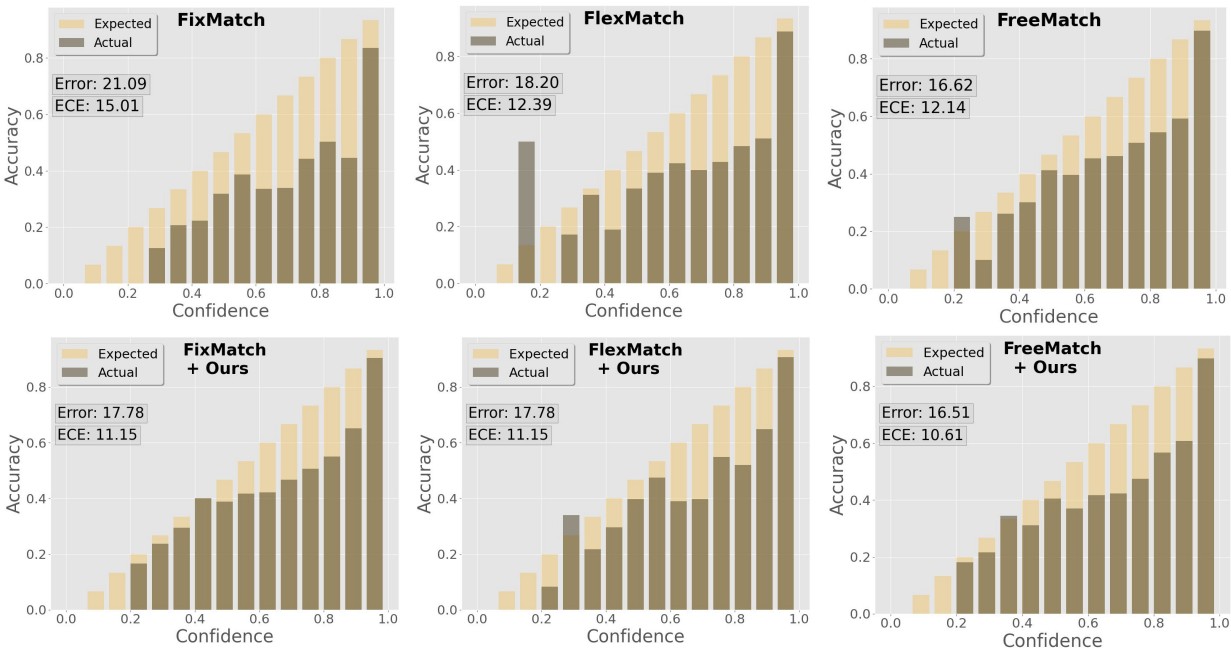

Figure 12: Reliability plots on CIFAR-100(400) for the original (*top*) and our (*bottom*) versions of the three pseudo-label SSL methods selected.

