# OpenReview forum: "Do not trust what you trust: Miscalibration in Semisupervised Learning"
_TMLR — Accepted by TMLR_

### Review · Reviewer_2xis · 2024-05-27

**Summary Of Contributions:**

The authors focus on addressing miscalibration in state-of-the-art semi-supervised learning (SSL) methods. They empirically demonstrate that these methods are significantly miscalibrated due to the minimization of the min-entropy term, which leads to overconfident and often incorrect predictions. They introduce a simple penalty term that enforces low logit distances in predictions on unlabeled samples, preventing overconfidence and improving calibration. Comprehensive experiments across various benchmarks show that this solution not only enhances calibration performance but also improves overall accuracy. The proposed method outperforms existing calibration techniques and proves effective in both standard and long-tailed classification tasks, offering a robust enhancement to SSL models.

**Audience:**

Yes

**Claims And Evidence:**

Yes

**Requested Changes:**

Please try to have some discussion with the weaknesses parts.

**Strengths And Weaknesses:**

Strengths
- Innovative Identification of Miscalibration Issues: The paper provides a thorough empirical and theoretical analysis of miscalibration in pseudo-labeling SSL methods, identifying the minimization of the min-entropy term as a key cause of overconfident predictions.

- Effective Solution with Simple Implementation: The proposed penalty term is straightforward to implement and effectively addresses the identified miscalibration issue, making it a practical addition to existing SSL frameworks.

- Comprehensive Experimental Validation: Extensive experiments across multiple benchmarks and tasks demonstrate the robustness and generalizability of the proposed solution, showing improvements in both calibration and accuracy.

Weaknesses:
- Focus on Hard Pseudo-labels: The primary focus is on SSL methods using hard pseudo-labels. While there is some discussion on methods using soft pseudo-labels, a more detailed analysis and comparison in this area could provide additional insights.

- Scalability and Computational Cost: The paper does not extensively discuss the computational cost and scalability of the proposed method, particularly for very large datasets or models. Including this analysis would help readers understand the practical implications of adopting the solution in real-world scenarios.

---

### Review · Reviewer_FYCB · 2024-06-11

**Summary Of Contributions:**

The paper addresses the issue of miscalibration in semi-supervised learning (SSL), showing that these models often become overly confident due to the minimization of min-entropy on unlabeled data, which is different from standard fully-supervised training. The authors leverage a margin-based label smoothing technique to mitigate overconfidence during training, which effectively improves the calibration without compromising the discriminative performance of SSL models across various image classification benchmarks.

**Audience:**

Yes

**Broader Impact Concerns:**

No need for Broader Impact Statement.

**Claims And Evidence:**

Yes

**Requested Changes:**

1. Adding discussion that acknowledges the existing studies in this area, highlighting the differences between your research and their work;

2. Adding existing methods for comparison in experiments;

3. Adding experiments or visualizations to illustrate the disparency of ECE for supervised training with and without consistency loss.

4. Adding more experiments for a more common experiment setup;

5. Properly address any other aforementioned issues.

**Strengths And Weaknesses:**

**Strengths:**

1. This paper studied a vital yet often neglected problem: probability calibration in SSL;

2. The insight provided in this paper is very interesting; the authors proposed that the SSL model tends to be more significantly miscalibrated compared to the fully supervised learning and attribute the potential cause as the different entropy-minimization scheme, which appears to be a very novel perspective;

3. The proposed calibration techniques do not require a labeled validation set to calculate ECE, which is very practical under the SSL setting.

**Weakness:**

1. Some highly relevant works on the importance of SSL and probability calibration are not included and discussed [1,2], all of which should be added for comparison in the experiment section.

2. Some claimed contributions overlap with existing works; specifically, the empirical demonstration of SSL is mis-calibrated and is not new, which has already been extensively discussed in existing works [1, 2].

3. The attribution on the cause of miscalibration in SSL lacks further support; if the consistency loss on unlabeled data is the cause of more severe miscalibration, this should be demonstrated by the disparency of calibration error between standard training with and without consistency loss.

4. As an empirical study, authors should consider adding experiments on more common settings, i.e., using WRN as most of the other works.

5. While I believe this paper is very insightful, I have doubts about the fundamental cause for the more significant miscalibration in SSL. Specifically, the authors believed it is because the loss is minimizing the min-entropy over the unlabeled data. **However, if the pseudo-labels are correct, the min-entropy of $\mathcal{D}_{U^{''}}$ has no difference to the standard CE loss,** except for the strong data augmentation. In other words, you can equivalently translate the CE on labeled data into the same decomposition, which contradicts the authors' hypothesis that SSL can lead to more severe miscalibration than fully supervised learning.

6. The solution seemed to be straightforwardly borrowed from existing works, undermining this work's technical contributions.

[1] "In Defense of Pseudo-Labeling: An Uncertainty-Aware Pseudo-label Selection Framework for Semi-Supervised Learning." ICLR, 2020.

[2] "On the Importance of Calibration in Semi-supervised Learning." arXiv, 2022.

---

### Review · Reviewer_RqMG · 2024-07-10

**Summary Of Contributions:**

This work studies calibration in semi-supervised learning based on pseudo-labels, where a supervised classification loss is combined with a pseudo-label loss on unlabeled samples. The contributions of this work are primarily empirical, with numerical experiments conducted with FixMatch and variants (FlexMatch, FreeMatch, and an ablation on SimMatch) on CIFAR, STL, and EuroSAT using a ViT-Small.

### Observations:
This work finds:
1. The expected calibration error (ECE) of FixMatch and variants is poor compared to a supervised baseline.
2. Hard pseudo-labels encourage the model to make confident predictions for individual samples; specifically, for images in the mini-batch  with the same argmax class prediction between weakly and strongly augmented views, the cross-entropy simply becomes an entropy minimization term (min-entropy).
3. More than 80% of the unlabeled samples have the same argmax class prediction between weakly and strongly augmented views at some point early in training; which the authors take to mean that semi-supervised learning using pseudo-labels can be approximated by a supervised loss and entropy minimization term.
4. The logits produced by FreeMatch are less concentrated and tend to have higher magnitude than those of a supervised learning baseline.

### Proposed solution:
Based on these observations, this work proposes to regularize FixMatch and related methods with a logit distance penalty, that encourages the pair-wise logit differences between classes to be **smaller** than a pre-specified margin (this is computed on a per-instance basis) to reduce overconfident predictions. Empirically, the proposed regularizer is reported to improve top-1 prediction accuracy and expected calibration error.

**Audience:**

Yes

**Broader Impact Concerns:**

N.A.

**Claims And Evidence:**

No

**Requested Changes:**

* On the claim that semi-supervised learning using pseudo-labels can be approximated by a supervised loss and an entropy minimization term; please consider running an ablation using only the third term in equation (6), while zeroing out the second term.
* Please clarify how you handle the fact that the margin penalty is not differentiable due to the strict max term.
* Please include SimMatch in all the tables/figures (not just Table 5) and expand the evaluation to the other datasets you consider.
* Please conduct ablations using various temperature parameters in the softmax (rather than the proposed penalty constraint).

**Strengths And Weaknesses:**

**Strengths:**
* The considered problem setting is interesting and broadly relevant for the TMLR community.
* The work is reasonably clear and easy to understand, and the figures are relatively well designed and make for a pleasant read.
* The proposed method does appear to improve performance on the considered tasks, both in terms of discriminative ability and calibration.

**Weaknesses:**

I have some concerns regrading the proposed solution and some of the claims made in this work.
* logits are unnormalized, hence one can minimize the distance between them by simply scaling them all down by the same constant. Since the margin hyper-parameter is fixed, you can achieve the same effect just by using a temperature parameter in the softmax without needing to induce a constrained optimization problem.
* the margin penalty is not differentiable due to the $max_j\ (\ell_{i,j})$ term, how do you handle this?
* appropriately softening the pseudo-label would achieve a similar effect; this can be seen in Table 5, where the effects of the proposed margin constraint are harmful in the few-label setting (200) and marginally helpful in the many label setting (1000) when using soft pseudo-labels (SimMatch).
* On the claim that semi-supervised learning using pseudo-labels can be approximated by a supervised loss and an entropy minimization term: I am not sure there is evidence to support this claim based on your exposition. Specifically, even if the third summation in equation (6) has more terms than the second term, the second term (consistency) may still provide important gradient signal. I expect that you will probably obtain notably degraded performance if you ran an ablation using only a supervised loss and an entropy minimization regularizer.

---

> ### Author Response · Authors · 2024-07-18
>
> Dear Reviewer,
>
> Thanks for your very constructive feedback! While we are working on addressing the raised concerns, we would like to better understand the experiments requested. In particular, we would like to request further details on the ablation experiments using various temperature parameters to scale the logits. Do you want us to report the post-training results for the different original methods (FixMatch, FlexMatch and FreeMatch) with various temperature scaling parameters? Or do you want that we train these methods (following equation 4) across different values of temperature scaling?
>
> Thank you

---

> ### Author Response · Authors · 2024-07-22
> **Response to the Reviewer RqMG (1/2)**
>
> 1. We would like to thank the reviewer for this valuable insight. We have modified this claim to better align with the theoretical and empirical analysis provided in this work. More concretely, in the revised version, we state that *we argue that the training of SSL methods based on pseudo-labels is equivalent to having a supervised term coupled with a pseudo cross-entropy on a small subset of unlabeled samples and a pseudo-entropy regularization loss that minimizes the min-entropy of the predictions from most unlabeled samples.* Note that this modification does not change our observations or the conveyed message. Since we have modified our claim, we deem the requested experiments unnecessary.
> 2. We would like to address this concern as :
> **Differentiability analysis:** The inner term $\max_j (l_{i,j})$ (which takes the maximum over all logits $(l_{i,j})$ for sample $i$) is piecewise linear but not differentiable at points where the maximum switches from one logit to another. In particular, if we consider two logits, $l_{i,1}(\theta)$ and $l_{i,2}(\theta)$, which depend on $\theta$, if there is a value $\theta_0$, where $l_{i,1}(\theta_0) = l_{i,2}(\theta_0)$, then, at point $\theta = \theta_0$, the function $\max_j(l_{i,j})$ switches from $l_{i,1}$ to $l_{i,2}$. Thus, at point $\theta = \theta_0$ the function is not differentiable.
> **Practical implications:** Taking into account that the $\max$ function is not differentiable only at the points where two (or more) logit values for a given sample $i$ are exactly the same, the amount of classes that each dataset contains, and the double precision numbers (16 digits) used in PyTorch, reaching this situation (two logits corresponding to the max value on a given sample) is virtually impossible. Thus, this function can be accommodated into standard stochastic gradient descent, not posing any additional challenge from an optimization standpoint. Please note that other non-differentiable functions, such as max-pooling and ReLU, are commonly used, yet this does not produce any problems with the optimization.
> 3. We would like to clarify the use of SimMatch in our work. As stated in the original version of the manuscript (please refer to the *Can we add our calibration strategy to other methods?* section), **SimMatch is used as a use-case to evaluate our approach in methods where our formulation does not necessarily hold**. In particular, for approaches that resort to soft pseudo-labels (such as SimMatch), we cannot strictly apply the reasoning that motivates the current work (i.e., Eq. 5 and Eq. 6) due to the use of pseudo-labels instead of hard-labels. We stress that this is highlighted in this section, and we further emphasize that we refrain from making claims regarding the benefits of our strategy in these soft pseudo-labels based approaches. Thus, we believe that adding systematically SimMatch in every Table and Figure would distort the conveyed message as well as the whole narrative of the current work, for the reasons given above.

---

> > ### Author Response · Authors · 2024-07-22
> > **Response to the Reviewer RqMG (2/2)**
> >
> > 4. Applying temperature scaling ($p_k = \frac{\exp^{l_k / \tau}}{\sum_{j=1}^K \exp^{l_j / \tau}}$) can indeed modify the shape of the softmax distribution, leading to smoother distributions when $\tau > 1$. Nevertheless, compared to the proposed solution, scaling the softmax values during training (before using the softmax probabilities in the cross-entropy loss) presents several drawbacks. **1.) No mechanism to control the importance of softened distributions:** Systematically scaling the softmax predictions might be suboptimal in several cases, such as: reduction of the gradient magnitude, loss of sparsity in the predictions, impaired learning of hard examples, or even magnifying the miscalibration issue, as strong softmax smoothing can lead to meaningless uniform distributions. While applying temperature scaling (TS) on the softmax values will always introduce a risk of these drawbacks appearing, the use of the proposed constrained term acts as a regularizer of the main objective, whose importance can be controlled by a blending hyperparameter $\lambda$, which introduces more flexibility during training. **2) Difficulty in finding the optimal scaling factor:** Softmax distributions at the beginning of the training (mostly uniform) are very different from those observed typically when the network converges (peaky distributions with one category dominating the softmax vector). Thus, the scaling factor to control the logit distance should also evolve during training (e.g., at the beginning, one might not want to scale the softmax at all, as this could magnify all the problems presented in the previous point). For instance, a scaling value of $2$ does not result in the same logit distances (or their equivalent softmax predictions) on an almost uniform distribution as it does on a peaky distribution. On the other hand, the proposed penalty acts on the absolute logit values, enforcing the same constraint across the training (i.e., just penalize those predictions whose logit distances are longer than the given margin), which is easier to adjust. **3) Different behavior:** Based on the previous point, at the beginning of the training, when the logit distribution is more uniform, the penalty term will not apply to most predictions, allowing the cross-entropy, pseudo cross-entropy, and min-entropy terms to guide the training. However, it will start to increasingly act as a regularizer as logit distributions become sharper, controlling the logits' magnitudes from becoming larger and larger, but allowing their distances to be large enough to have discriminative power. On the other hand, scaling the softmax from the beginning will reduce the softmax values of all the samples systematically, which may impede the network from properly learning. Thus, we can argue that the proposed constrained term offers more control over the logit/softmax distributions than simply scaling the softmax predictions, more flexibility as it offers a mechanism to control the importance of the constraint, and a different behavior, particularly at the beginning of the training.
> > Presently we are conducting experiments with temperature scaling during training which we shall conclude soon.

---

> > > ### Author Response · Authors · 2024-07-24
> > > **Response to the Reviewer RqMG - Experiments**
> > >
> > > We have now concluded our experiments with the following results with 4 different temperature settings (2, 5, 10, and 20) for the CIFAR-100 dataset with both settings of labeled data:
> > >
> > >
> > > | # Labeled Samples | Temperature         | ECE                 | Error               |
> > > |-------------------|---------------------|---------------------|---------------------|
> > > | **200**           |                     |                     |                     |
> > > |                   | 2                   | 26.63±0.93          | 29.53±1.07          |
> > > |                   | 5                   | 26.58±0.31          | 27.93±0.46          |
> > > |                   | 10                  | 27.46±0.67          | 28.07±0.70          |
> > > |                   | 20                  | 27.55±0.92          | 27.98±1.03          |
> > > | **400**           |                     |                     |                     |
> > > |                   | 2                   | 17.49±1.23          | 20.43 ±1.04              |
> > > |                   | 5                   | 18.32±0.90          | 19.69±1.11          |
> > > |                   | 10                  | 18.85±0.33          | 19.48±0.34          |
> > > |                   | 20                  | 18.78±0.52          | 19.10±0.51          |
> > >
> > > We also add logit distribution plots in our revised manuscript (Figure 7) where it can be observed that the impact on the logit distribution of our method, and systematically applying temperature scaling to all samples, is different, which empirically supports our hypothesis regarding their different behaviour.

---

### Decision · Action_Editor_WSBw · 2024-08-26

**Recommendation:** Accept as is

**Comment:**

No further comments beyond those in claims/evidence and audience.

**Audience:**

Yes, it would be of interest to anyone developing or applying consistency regularization/pseudo-labeling-based semi-supervised learning. Such researchers are in TMLR's audience.

**Claims And Evidence:**

Reviewers generally felt convinced by the claims in the paper and the experimental setting. Overall, the experimental results are a bit mixed, but there is some signal that the proposed method does what it is claimed to. One reviewer questioned whether the paper had identified the true underlying source of miscalibration in modern SSL methods. Having looked through it, I agree that it's not entirely clear that the minimization of the min-entropy is the *only* or even primary cause of miscalibration in SSL, but I don't think that's the authors' claim - they simply point to it as a possible cause and present a possible mitigation strategy that seems to help somewhat. I think the paper can provide a useful bit of further exploration into this phenomenon.